JCB Journal of Cell Biology

# NASP functions in the cytoplasm to prevent histone H3 aggregation during early embryogenesis

Mohit Das[1]*, Eli Coronado-Chavez[1]*, Anusha D. Bhatt[2], Reyhaneh Tirgar[1], Amanda A. Amodeo[2], and Jared T. Nordman[1]

**From their molecular birth until their incorporation into chromatin, histones are bound by specific chaperones that serve unique functions in histone trafficking, stability, and chromatin deposition. The H3-specific chaperone NASP binds directly to H3 and is required to prevent degradation of soluble H3 in vivo. Where NASP functions and how NASP affects H3 dynamics and stability are unknown. Using the *Drosophila* early embryo as a model system to understand NASP function in vivo, we show that NASP does not directly affect H3 nuclear import or export rates. Rather, reduced H3 levels in NASP-deficient embryos indirectly affect nuclear import and the amount of H3 deposited into chromatin. Crucially, we find that cytoplasmic NASP prevents H3 aggregation in vivo and that H3 aggregation and degradation are developmentally separable events. Thus, we propose the main function of NASP in vivo is to prevent H3 aggregation, thereby indirectly protecting H3 from degradation.**

## Introduction

To package the eukaryotic genome, DNA is wrapped around an octamer of histones to form nucleosomes (Kornberg, 1974; Noll and Kornberg, 1977; Arents and Moudrianakis, 1993; Khorasanizadeh, 2004). Histones are not only responsible for condensing DNA to fit within the nucleus, but they also regulate access to the genetic information by controlling DNA accessibility (Noll and Kornberg, 1977; Khorasanizadeh, 2004; Talbert and Henikoff, 2017; Bannister and Kouzarides, 2011). Histones affect nearly every aspect of chromatin metabolism, and their synthesis must be carefully regulated (Khorasanizadeh, 2004; Talbert and Henikoff, 2017; Bannister and Kouzarides, 2011; Kornberg and Lorch, 2020; Meeks-Wagner and Hartwell, 1986). Histone limitation can increase sensitivity to DNA damage and global increases in transcription (Kornberg and Lorch, 2020; Gunjan and Verreault, 2003; Celona et al., 2011). Conversely, oversupply of histones causes genomic instability and chromosome loss in budding yeast (Meeks-Wagner and Hartwell, 1986; Singh et al., 2010; Reis and Campbell, 2007). Excess histones can nonspecifically bind to DNA and RNA, which has the potential to impact replication, transcription, and translation (Meeks-Wagner and Hartwell, 1986; Singh et al., 2010; Mei et al., 2017). To circumvent potential histone toxicity in somatic cells, histone production peaks in S phase when the demand is the highest and soluble (non-chromatin–bound) histone pools are held to <1% of the total histone supply (Osley, 1991; Zhao et al., 2000; Oliver et al., 1974; Bonner et al., 1988; Gunjan et al., 2006; Marzluff et al., 2008).

Embryonic development in flies, fish, and frogs requires a massive and immediate supply of histones to sustain the rapid nuclear divisions typical of early embryonic development (Horard and Loppin, 2015; Woodland and Adamson, 1977; Joseph et al., 2017). The early *Drosophila* embryo transforms from a single nucleus to 4,000–6,000 nuclei in only 2 h (Yuan et al., 2016; Vastenhouw et al., 2019). The first 13 nuclear cycles are rapid, synchronous, and occur in a syncytium. During the first 13 nuclear divisions, embryogenesis is fueled by maternal deposits of proteins and RNAs, including histones (Ambrosio and Schedl, 1985; Walker and Bownes, 1998; Farrell and O'Farrell, 2014; Yuan et al., 2016; Vastenhouw et al., 2019). The mid-blastula transition occurs at nuclear cycle 14, which coincides with cell cycle slowing, cellularization, and the onset of zygotic transcription (Yuan et al., 2016; Vastenhouw et al., 2019; Farrell and O'Farrell, 2014; Chen et al., 2013). In contrast to somatic cells, the vast majority of histones are soluble in the earliest stages of embryogenesis, and the soluble supply slowly decreases throughout embryogenesis (Horard and Loppin, 2015).

It is a significant challenge to measure the rate of nuclear import and export of histones in somatic cells. This is because the rate of histone nuclear import is faster than the time it takes for a fluorescent reporter such as GFP to fold and mature (Bonner et al., 1988; Ruiz-Carrillo et al., 1975). Given histones are maternally deposited in early *Drosophila* embryos, nuclear import and export rates of histones have been measured and

---

[1]Department of Biological Sciences, Vanderbilt University, Nashville, TN, USA; [2]Department of Biological Sciences, Dartmouth College, Hanover, NH, USA.

*M. Das and E. Coronado-Chavez contributed equally to this paper. Correspondence to Jared T. Nordman: jared.nordman@vanderbilt.edu

Reyhaneh Tirgar's current affiliation is Laboratory of Cellular and Developmental Biology, National Institute of Diabetes and Digestive and Kidney Diseases Intramural Research Program, National Institutes of Health, Bethesda, MD, USA.

characterized in early embryos (Shindo and Amodeo, 2019). For the replication-dependent histone H3.2, the rate of nuclear import decreases every nuclear cycle due to the depletion of maternally provided H3.2 that occurs upon each nuclear division (Shindo and Amodeo, 2019; Shindo and Amodeo, 2021). The cytoplasmic pool of H3.2 becomes limiting in nuclear cycle 13, leading to a reduction of chromatin-associated H3.2 (Shindo and Amodeo, 2019). This reduction is compensated by increased incorporation of the replication-independent H3 variant, H3.3, into chromatin starting in nuclear cycle 13 (Shindo and Amodeo, 2019). Thus, the import kinetics of H3 and its variants are concentration dependent and developmentally regulated.

To maintain the large maternal stores of histones and to suppress the potential toxicity associated with excess histones, embryos employ sophisticated mechanisms to store and regulate the availability of histones through early development. Histone chaperones are a class of proteins that associate with histones and are crucial for histone stability and histone metabolism (Pardal et al., 2019; Gurard-Levin et al., 2014). While few chaperones bind to all histone subunits, most histone chaperones are specific to H2A–H2B or H3–H4 (Hammond et al., 2017; Natsume et al., 2007; Elsässer et al., 2012; Ramos et al., 2010). In Drosophila embryos, the histone chaperone Jabba binds to H2A, H2B, and H2Av and sequesters them to lipid droplets, protecting them from degradation (Li et al., 2012). Nuclear autoantigenic sperm protein (NASP) is a histone chaperone that binds to H3–H4 reserves in both mammalian cells and Drosophila embryos and protects them from degradation (Tirgar et al., 2023; Cook et al., 2011; Campos et al., 2015). Unlike Jabba, NASP is a maternal effect gene, and embryos laid by NASP-mutant mothers have severely reduced hatching rate (Li et al., 2012; Tirgar et al., 2023; Zhang et al., 2018).

Humans express two NASP isoforms generated by alternative splicing, testicular NASP and somatic NASP, whereas Drosophila encodes a single NASP isoform (Richardson et al., 2000; Tirgar et al., 2023). NASP binds directly to H3 through evolutionarily conserved tetratricopeptide repeat domains and the NASP–H3 interaction has been extensively characterized (Nabeel-Shah et al., 2014; Bowman et al., 2016; Liu et al., 2021). Understanding how NASP controls H3 dynamics in vivo, however, is less well understood. Biochemical work in mammalian cells has identified a number of NASP–H3-containing complexes derived from cytosolic extracts (Campos et al., 2015; Tagami et al., 2004; Pardal and Bowman, 2022). Subsequent work revealed that NASP localizes to the nucleoplasm and that NASP can rapidly diffuse out of the nucleus during preparation of cytosolic extracts (Apta-Smith et al., 2018). In cells depleted of NASP, soluble pools of H3 have been inferred to be degraded through chaperone-mediated autophagy (CMA), which occurs in the cytoplasm (Cook et al., 2011; Hormazabal et al., 2022). Taken together, it is unclear if NASP functions in the nucleus or cytoplasm to protect soluble H3 from degradation, if NASP has any direct impact on the nuclear import or export of H3, or if NASP directly protects H3 from degradation. In mammalian cells, it has been proposed that NASP could act as a nuclear receptor for H3, thereby preventing H3 from nuclear export (Apta-Smith et al., 2018). Given the caveats of measuring the import and export

dynamics of H3 in somatic cells, this model is difficult to test directly. Thus, it is still unclear how NASP functions in vivo to control the stability and trafficking of H3.

Previously, we demonstrated that the Drosophila homolog of NASP binds specifically to H3, H3.3, and H4 in both Drosophila oocytes and embryos (Tirgar et al., 2023). Furthermore, H3 and H4 were both degraded in embryos devoid of NASP, while H2A and H2B were unaffected (Tirgar et al., 2023). Similar to somatic mammalian cells, NASP functions to protect H3 and H4 from degradation in Drosophila embryos and oocytes. Here, we leverage the Drosophila developmental system to directly measure how NASP affects the import and export dynamics of H3. We show that NASP has no direct effect on the nuclear import or export rates of H3.2 or H3.3. In embryos devoid of NASP, histone import rates are reduced, but this reduction matches the expected rate based on reduced H3 protein levels. Interestingly, embryos devoid of NASP have approximately half the amount of H3 in chromatin relative to NASP-proficient embryos, yet these embryos progress through nuclear cycles 10–13 with little cell cycle slowing or mitotic defects. We show that NASP is largely nucleoplasmic in Drosophila embryos but functions in the cytoplasm to stabilize H3. Finally, we show that NASP functions in the cytoplasm to prevent H3 aggregation. Thus, H3 aggregation occurs prior to degradation in vivo. Altogether, we propose that the main function of NASP in vivo is to protect H3 from aggregation and that cytoplasmic aggregates of H3 formed in the absence of NASP are targeted for degradation.

## Results

### H3 nuclear import is independent of NASP

Understanding how NASP functions to control nuclear import of its direct binding partner H3 has been difficult for two reasons. First, in somatic cells, the vast majority of cellular histones are chromatin bound, and <1% of total H3 is soluble (non-chromatin bound) (Loyola et al., 2006). This limits the amount of H3 for interaction and nuclear import studies. Second, there is a kinetic barrier in studying histone transport since the rate of nuclear histone import is faster than the folding and maturation kinetics of fluorescent proteins (Reits and Neefjes, 2001; Lukyanov et al., 2005; Ishikawa-Ankerhold et al., 2012). To overcome the challenges of histone tracking in somatic cells, we utilized the Drosophila early embryo as a model system. Early Drosophila embryos are stockpiled with maternal reserves of RNA and proteins, and a majority of the maternal deposited histones are soluble at the earliest stages of embryogenesis (Horard and Loppin, 2015). Thus, early embryos have large soluble pools of translated histones, serving as an excellent system to study H3 dynamics (Horard and Loppin, 2015; Shindo and Amodeo., 2019; Shindo and Amodeo., 2021).

To test if H3.2 import is dependent on NASP, we utilized the photo-switchable fluorescent reporter Dendra2 tagged to H3.2 for live imaging of early embryos laid by WT or null NASP-mutant mothers. NASP is not essential for viability but loss of NASP function results in female sterility. A subset of embryos laid by NASP-mutant mother can progress through embryogenesis (Tirgar et al., 2023). The tagged H3.2-Dendra2 is

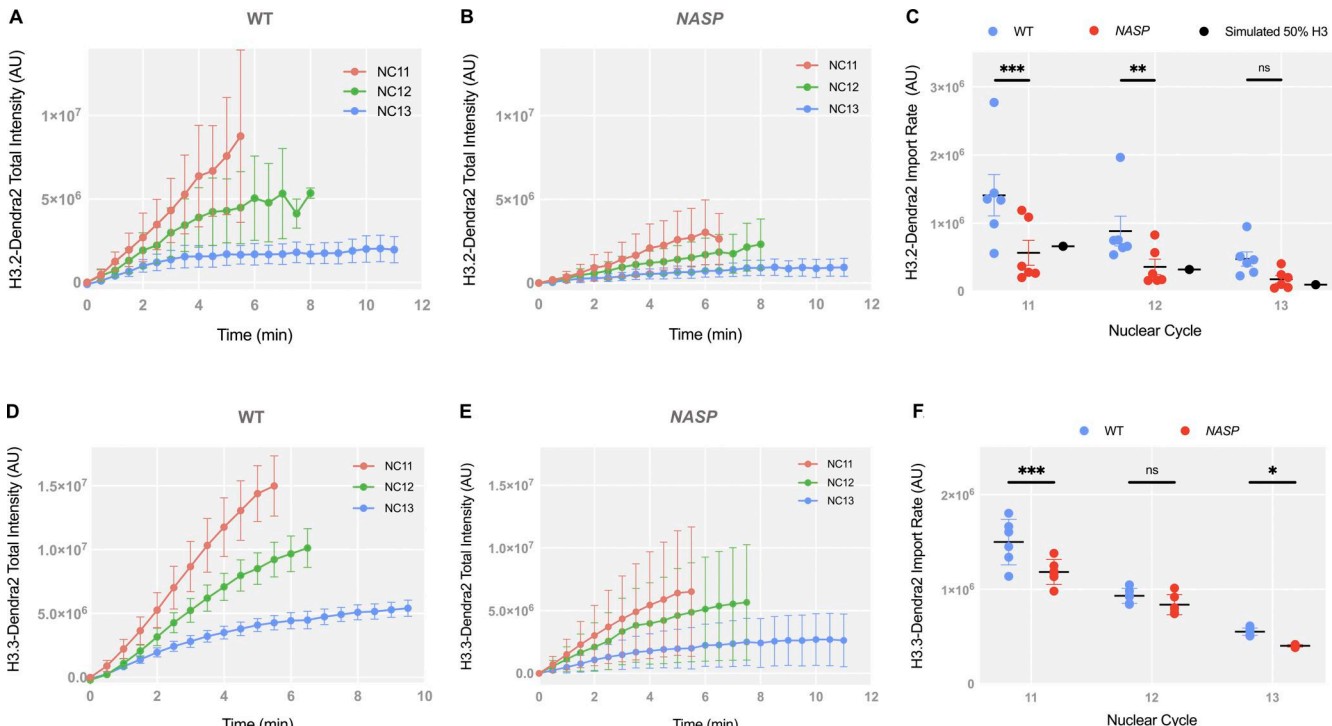

Figure 1. **H3 nuclear import is independent of NASP. (A)** Total intensity curves of H3.2-Dendra2 for nuclear cycles (NC) 11–13 in WT embryos. **(B)** Total intensity curves of H3.2-Dendra2 for nuclear cycles (NC) 11–13 in NASP-deficient embryos. **(C)** Nuclear import rates of H3.2-Dendra2 in WT (blue) and NASP-deficient (red) embryos. Simulated import rate for H3.2 at 50% WT levels (black). N = 6 biological replicates. Data represent the mean ± standard deviation (SD). Statistical difference between groups were analyzed by employing a two-way ANOVA Tukey's multiple comparison test between the WT and *NASP* data points. **(D)** Total intensity curves of H3.3-Dendra2 for nuclear cycles (NC) 11–13 in WT embryos. **(E)** Total intensity curves of H3.3-Dendra2 for nuclear cycles (NC) 11–13 in NASP-deficient embryos. **(F)** Nuclear import rates of H3.2-Dendra2 in WT (blue) and NASP-deficient (red) embryos. N = 6 experimental replicates. Data represent the mean ± standard deviation (SD). Statistical difference between groups were analyzed by employing a two-way ANOVA Tukey's multiple comparison test between the WT and *NASP* data points. In the figure panels, ns represents nonsignificant difference while *P < 0.05, **P < 0.01, and ***P < 0.001.

expressed from a single copy of the histone gene locus that includes all promoters, coding sequences and UTRs to ensure endogenous regulation (Marzluff et al., 2008). The H3.2-Dendra2 reporter undergoes posttranslational modifications in vivo and interacts with endogenous cytoplasmic-binding partners (Li et al., 2012; Hödl and Basler., 2012; Kimura and Cook, 2001). To measure the rate of H3.2 nuclear import, we performed live imaging of non–photo-switched (green) H3.2-Dendra2 in 30″ intervals and calculated the total nuclear intensities across nuclear cycles 11–13 in embryos laid by WT or *NASP*-mutant mothers (Video 1). Given the differences in histone concentration between embryos laid by WT or *NASP*-mutant mothers, we measured the absolute levels of H3.2-Dendra intensities throughout nuclear cycles 11–13 (Fig. 1, A and B). The nuclear import rates were measured by calculating the initial slopes of nuclear H3.2-Dendra2 intensities for each nuclear cycle 11–13 (Shindo and Amodeo, 2019). We found that there is a significant decrease in the rate of H3.2-Dendra2 nuclear import for nuclear cycles 11 and 12 in embryos laid by *NASP*-mutant mothers compared with embryos laid by WT mothers (Fig. 1, A–C). While the rate of nuclear cycle 13 was also reduced, it did not reach the level of significance (Fig. 1, A–C). Similar results were obtained when normalizing the total nuclear intensities to the maximum nuclear H3.2-Dendra2 intensity in nuclear cycle 11 immediately

prior to nuclear envelope breakdown (Fig. S1, A–C). NASP also binds to the non-replicative H3 variant H3.3 (Shindo and Amodeo, 2019; Tirgar et al., 2023). Thus, we tested if the nuclear import of H3.3 is dependent on NASP using the same live-imaging technique for embryos tagged with H3.3-Dendra2 (Shindo and Amodeo, 2019). We observed a similar decrease in the nuclear import rate of H3.3-Dendra2 in embryos laid by *NASP*-mutant mothers compared with embryos laid by WT mothers (Fig. 1, D–F and Fig. S1, D–F).

The rate of H3.2 import is dependent on the total H3.2 concentration (Shindo and Amodeo, 2021). While the nuclear import rate of H3.2-Dendra2 was reduced in the absence of NASP, H3.2 protein levels are reduced ∼50% in embryos laid by *NASP*-mutant mothers compared with embryos laid by WT mothers (Tirgar et al., 2023). Therefore, it is possible that the decreased H3.2 import rate in embryos laid by *NASP*-mutant mothers is solely due to reduced H3.2 protein levels. To test this possibility, we used an established integrative model to simulate the rate of H3.2 nuclear import in a concentration-dependent manner (Shindo and Amodeo, 2021). The nuclear import of H3 follows Michaelis–Menten kinetics and is sensitive to H3 concentration (Shindo and Amodeo, 2021). We simulated H3.2 import rates with a 50% reduction of H3.2 levels to align with our experimentally measured values by mass spectrometry and western

blotting (Tirgar et al., 2023) (Fig. S1, G and H). Our experimentally derived nuclear import data align with simulated import rates with a 50% reduction in H3.2 concentration using either unnormalized values (Fig. 1 C) or values normalized to nuclear cycle 11 (Fig. S1 C). Taken together, we conclude that NASP does not directly affect the nuclear import of H3, and the decrease in nuclear import rate of H3.2 in embryos laid by *NASP*-mutant mothers is solely attributed to reduced H3.2 levels.

### NASP-deficient embryos have less nucleoplasmic and chromatin-associated H3

It has been suggested that NASP could function as a nuclear receptor for H3.2, thereby retaining H3.2 in the nucleus and preventing its export (Pardal and Bowman, 2022). The measured rate of H3.2 export from the nucleus in WT embryos is negligible, so if there is a factor preventing H3.2 export from the nucleus, NASP is an ideal candidate (Pardal and Bowman, 2022). We tested if NASP prevents H3.2 export by photo-converting the H3.2-Dendra2 reporter in nuclear cycle 12 and measuring the intensity of photoconverted H3.2-Dendra2 over the course of nuclear cycle 12 (Fig. 2 A). There was no significant change in the nuclear intensity of photoconverted H3.2-Dendra2 prior to nuclear envelope breakdown for embryos laid by WT or *NASP*-mutant mothers, indicating that NASP does not affect export of H3.2 (Fig. 2 B and Fig. S1 J). There is a significant reduction in H3.2-Dendra2 levels upon nuclear envelope breakdown in embryos laid by WT mothers due to the loss of non-chromatin–bound nucleoplasmic H3.2-Dendra2 (Fig. 2 B) (Shindo and Amodeo, 2019). In contrast, there was only a slight reduction in H3.2-Dendra2 upon nuclear envelope breakdown in embryos laid by *NASP*-mutant mothers (Fig. 2 B). Similar results were observed for H3.3 (Fig. 2 C and Fig. S1 K). Consistent with the reduced levels of total H3.2 and H3.3 in embryos laid by *NASP*-mutant mothers, these data reveal that the level of soluble nucleoplasmic H3.2 and H3.3 is reduced in *NASP*-deficient embryos. We conclude that NASP indirectly reduces both the nuclear import rate and the soluble supply of both H3.2 and H3.3 in the nucleoplasm.

Given the reduced import rate and nucleoplasmic concentration of H3.2 in embryos devoid of NASP, we wondered if this would lead to a reduction in the amount of chromatin-associated H3.2. To test this, we compared the total mitotic H3-Dendra2 intensities in embryos laid by WT and *NASP*-mutant mothers. We measured the absolute intensity of H3.2-Dendra2 in metaphase nuclei across nuclear cycles 10–13 (NC10–13) for these embryos (Fig. 2 D). The total H3.2-Dendra2 intensity across NC10–NC13 was significantly reduced in embryos laid by *NASP*-mutant mothers relative to embryos laid by WT mothers (Fig. 2 E). The reduced levels of chromatin-bound H3.2 in embryos devoid of NASP are similar to the overall reduction in H3 in these embryos (Fig. 2 E) (Tirgar et al., 2023). It is possible that the H3.3 could replace H3.2 in NASP-deficient embryos to maintain a constant nucleosome:DNA ratio. Therefore, we also measured chromatin-bound H3.3 in embryos laid by WT and *NASP*-mutant mothers (Fig. 2 F). Similar to the overall reduction in H3.3 protein levels, we found that the H3.3 protein levels were also reduced in chromatin (Fig. 2 G). Thus, H3.3 does not

compensate for reduced H3.2 levels in chromatin in NASP-deficient embryos. Despite the differences in chromatin-bound H3, embryos devoid of NASP have only a modest effect on cell cycle length starting in nuclear cycle 13 (Fig. S1 I). We conclude that NASP indirectly affects the amount of H3.2 and H3.3 that gets incorporated into the chromatin during early nuclear divisions.

### H3 and NASP have different import kinetics

While H3 import kinetics are independent of NASP, it is possible that NASP and H3 form a complex in the cytoplasm that is imported into the nucleus. Biochemical data support the formation of a NASP–H3 complex in the cytoplasm; however, NASP dissociates from H3 upon the binding of importin-4 or importin-5 in mammalian cells (Pardal and Bowman, 2022). Localization studies, coupled with in vitro assays, suggest that NASP is nuclear-localized in mammalian cells and that NASP complexes with H3 in the nucleus (Pardal and Bowman, 2022; Apta-Smith et al., 2018). To date, there have been no studies that measure the localization and nuclear import kinetics of endogenous NASP and H3 to determine their nuclear import kinetics in vivo. Nor is it known if NASP localizes to the nucleus during *Drosophila* embryogenesis. Therefore, we used CRISPR-based mutagenesis to add a Dendra2 tag to the 3′ end of *NASP* at the endogenous locus, generating a C-terminally–tagged NASP-Dendra2 protein. Homozygous *NASP-Dendra2* flies were verified by sequencing and western blot (Fig. S2, A and B). Homozygous *NASP-Dendra2* females were fertile, indicating that the NASP-Dendra2 protein was functional. Live imaging of early embryos expressing NASP-Dendra2 revealed that NASP was largely nuclear localized, similar to NASP localization in *Drosophila* cultured cells and mammalian cells (Fig. 3 A and Video 2) (Tirgar et al., 2023; Apta-Smith et al., 2018). The NASP-Dendra2 intensity is nuclear during interphase and completely disperses upon nuclear envelope breakdown, while the H3.2-Dendra2 intensity persists throughout mitosis (Fig. 3 A). Thus, NASP is imported into the nucleus during early nuclear divisions and remains nucleoplasmic until it disperses upon nuclear envelope breakdown.

With NASP-Dendra2, we could now measure the nuclear import kinetics of NASP and H3 during the nuclear divisions of the early embryo. Absolute nuclear import rates are dependent on the concentration of the imported protein (Timney et al., 2006). Given the concentration difference of Dendra2-tagged H3.2 and NASP in early embryos, we cannot directly compare the absolute nuclear import rates for these two proteins. We could, however, measure and compare the relative nuclear import rate of NASP-Dendra2 and H3.2-Dendra across nuclear cycles 11–13 (Fig. 3, B and C). We observed that the nuclear import dynamics of NASP-Dendra2 are starkly different from H3.2 (Fig. 3 D). Relative nuclear import rates of H3.2-Dendra2 are reduced by half each nuclear cycle (NC11–13), consistent with exhaustion of the H3.2 pools as the embryo progresses through embryogenesis (Fig. 1 A and Fig. 3 C). The NASP-Dendra2 relative nuclear import rates, however, did not significantly change between nuclear cycles 11 and 12 (Fig. 3 B). Only in nuclear cycle 13 was there a decrease in NASP-Dendra2 import rate (Fig. 3, B and D). These kinetics were significantly different than those of

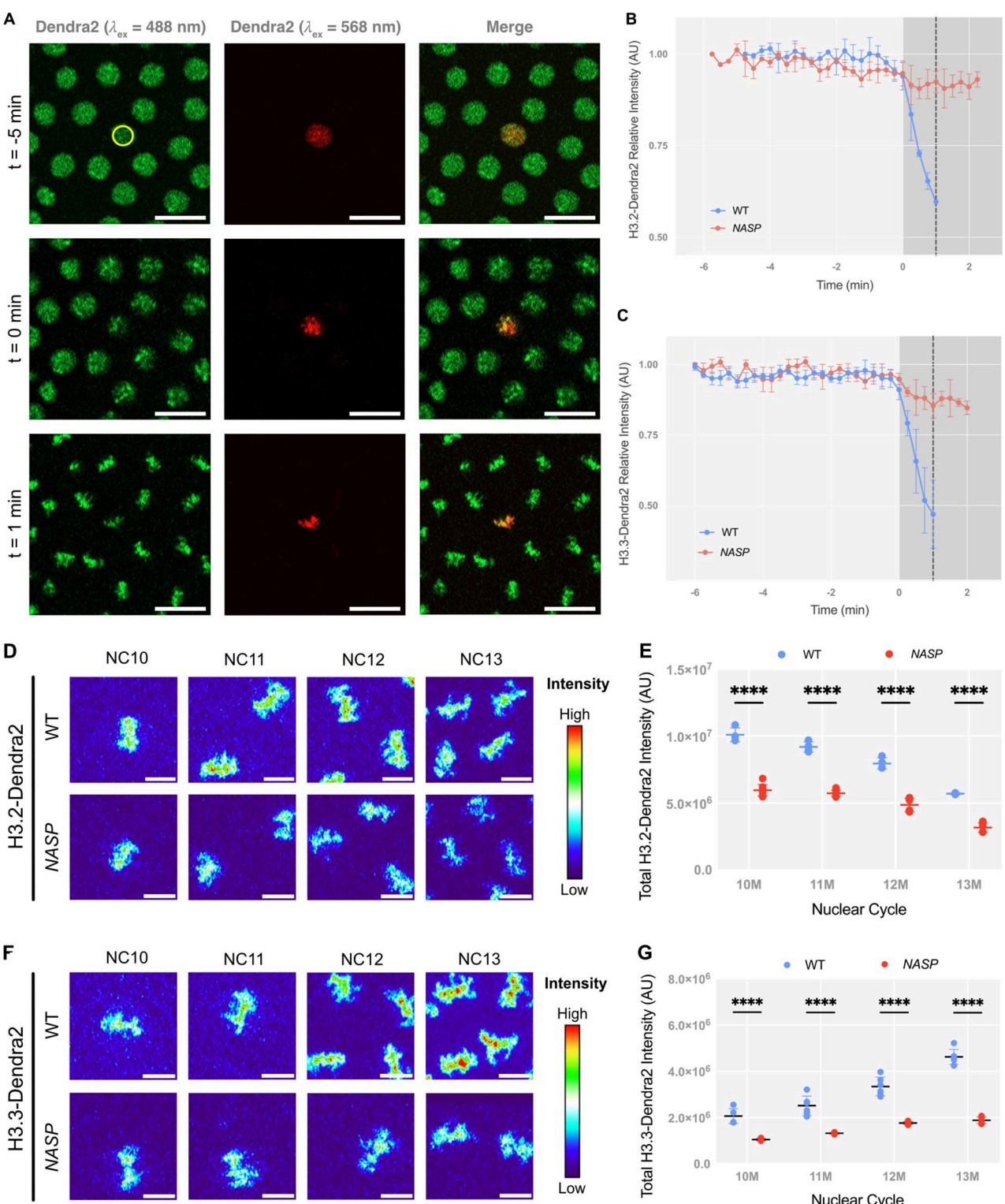

Figure 2. **NASP-deficient embryos have less nucleoplasmic and chromatin-associated H3. (A)** Representative images showing photoconversion of H3.2-Dendra2. Non–photo-switched Dendra2 is shown in green ($\lambda_{ex}$ = 488 nm), and the photo-converted Dendra2 is shown in red ($\lambda_{ex}$ = 568 nm). The yellow circle represents the nucleus irradiated by the 405 nm laser. Scale bar represents 20 μm. **(B)** Nuclear export measurement for H3.2-Dendra2 in WT and NASP-deficient embryos in NC12. Time point 0 represents Nuclear envelop breakdown (NEB). *N* = 3 biological replicates. Data represent the mean ± standard deviation (SD). **(C)** Nuclear export measurement for H3.3-Dendra2 in WT and NASP-deficient embryos in NC12. *N* = 3 biological replicates. Data represent the mean ± standard deviation (SD). **(D)** Maximum intensity projections of metaphase chromatin from WT (top) and NASP-deficient (bottom) embryos expressing H3.2-Dendra over NC10-13. Heat map created by pseudo-coloring images using nonlinear Thermal LUT on FIJI. Scale bar represents 10 μm. **(E)** Total intensities of H3.2-Dendra2 on mitotic chromatin of WT and NASP-deficient embryos for metaphase 10–13. *N* > 4 biological replicates. Data represent the mean ± standard deviation (SD). Statistical difference between groups were analyzed by employing a two-way ANOVA Sídák's multiple comparison test. **(F)** Maximum intensity

projections of metaphase chromatin from WT (top) and NASP-deficient (bottom) embryos expressing H3.3-Dendra over NC10-13. Heat map created by pseudo-coloring images using nonlinear thermal LUT on FIJI. Scale bar represents 10 μm. **(G)** Total intensities of H3.3-Dendra2 on mitotic chromatin of WT and NASP-deficient embryos for metaphase 10–13. *N* > 4 biological replicates. Data represent the mean ± standard deviation (SD). Statistical difference between groups was analyzed by employing a two-way ANOVA Sídák's multiple comparison test. In the figure panels, ****P < 0.0001.

H3.2-Dendra2 (Fig. 3 D). Subtracting the chromatin-bound values of H3.2 to measure the import rate of only the soluble fraction of H3.2 did not affect the import kinetics (Fig. S2 C). These observations show that H3.2 and NASP have different nuclear import kinetics during the nuclear divisions of the early embryo. Thus, in agreement with biochemical data, NASP and H3.2 are unlikely to be imported into the nucleus as a single complex and are rather imported independent of each other. Together with our data showing that the rate of H3 import or export is not dependent on NASP, we conclude that NASP has no direct effect on H3 import or export dynamics and that NASP is unlikely to act as a nuclear receptor for H3 during the nuclear divisions of the early embryo.

**NASP prevents H3 aggregation**

We previously showed that H3 is not degraded in *NASP*-mutant stage 14 egg chambers (Tirgar et al., 2023). While the total H3 levels are comparable in WT and *NASP*-mutant stage 14 egg chambers, H3 levels are reduced in 0–2 h embryos laid by *NASP*-mutant mothers (Tirgar et al., 2023). To test if degradation of H3 is tied to egg activation or the onset of embryogenesis, we crossed WT and *NASP*-mutant mothers to loss of function *twine*-mutant males to generate activated eggs that fail to progress into active embryos (Alphey et al., 1992). Twine is essential for the completion of meiosis, and *twine*-mutant males fail to make sperm (Alphey et al., 1992; Courtot et al., 1992; Edgar and Datar, 1996). Overnight collections of eggs laid by WT females crossed with *twine*-mutant males produce activated eggs that do not enter embryogenesis and thus do not have mitotic nuclei (Fig. S4, A and B). We crossed WT or *NASP*-mutant mothers to *twine*-mutant males and collected activated eggs. Western blot analysis of stage 14 egg chamber lysates from WT and NASP-deficient females reveal comparable levels of total H3, consistent with our previous findings (Fig. 4, A and B) (Tirgar et al., 2023). Activated eggs laid by *NASP*-deficient females, however, have significantly lower total H3 level relative to activated eggs laid by WT females (Fig. 4, A and B). Thus, the degradation of H3 is tied to egg activation and not fertilization or entry into embryogenesis. Further western blot analysis revealed that this reduction was specific to H3, as H2B levels remain unaffected in eggs laid by *NASP*-mutant mothers (Fig. 4 C and Fig. S3 C). Importantly, activated eggs do not contain mitotic nuclei, yet NASP is required to prevent H3 degradation in these eggs. Therefore, NASP can function in the cytoplasm to protect H3 from degradation upon egg activation.

Histones are highly prone to aggregation in vitro (Aragay et al., 1991). Furthermore, in *NASP*-mutant stage 14 egg chambers, there is a higher fraction of insoluble H3 as compared with the WT egg chambers (Tirgar et al., 2023). Therefore, the primary function of NASP during oogenesis could be to prevent H3 aggregation. To test this possibility, we adopted an aggregate

isolation assay used to isolate protein aggregates from metazoan tissue lysate (Fig. 5 A) (Chen et al., 2024). By using differential centrifugation, this protocol separates aggregates from larger complexes, including organelles, ribonucleoproteins, and condensates (Kryndushkin et al., 2013; Kryndushkin et al., 2017; Jain et al., 2016; Mitchell et al., 2013; Sheth and Parker, 2003; Wheeler et al., 2017; Ingolia et al., 2012). Stage 14 egg chambers were collected from WT and *NASP*-mutant females, lysed, and subjected to differential centrifugation to obtain a lysate for ultracentrifugation. These lysates were subject to ultracentrifugation to isolate protein aggregates (Fig. 5 A). To validate the assay in isolating protein aggregates from *Drosophila* egg chamber lysates, we used aggregate inducing amino acid analogue treatments as positive controls (Hare, 1970; Rosenthal 1977). WT ovaries were treated with increasing concentrations of canavanine (L-arginine analogue) for 3 h prior to aggregate isolation. We used whole ovaries to ensure amino acid analogs could permeabilize egg chambers. Western blot analysis of the fractions revealed a dose-dependent increase in aggregate formation at increasing levels of canavanine in both whole ovaries and *Drosophila* Schneider 2 (S2) cells (Fig. 5, B and C; and Fig. S4, C and D). A similar increase in aggregate formation was observed when WT ovaries were treated with azetidine-2-carboxylic acid (L-proline analogue) (Fig. S4, A and B) (Weids et al., 2016).

Western blot analysis of WT stage 14 egg chambers revealed that ~9.5% of H3 was present in the aggregate fraction (Fig. 5, D and E; and Fig. S5 A). The amount of H3 in the aggregate fraction of *NASP*-mutant stage 14 egg chambers, however, increased ~11-fold relative to 104.4% (Fig. 5, D and E; and Fig. S5 A). Aggregation was specific to H3, as there was no significant enrichment of H2B in aggregates derived from *NASP*-mutant egg chambers (Fig. 5, D and E). The decrease in H3 levels in *NASP*-mutant input fractions compared with WT input fractions is likely due to the altered solubility of H3 in the absence of NASP, causing loss of H3 in early centrifugation steps during aggregate isolation. Given that H3 degradation in the absence of NASP is tied to egg activation (Fig. 4 A), we next asked if the aggregate fraction of H3 was preferentially targeted for potential degradation in the absence of NASP. We collected activated eggs from WT and *NASP*-mutant mothers that were crossed to *twine*-mutant males and performed the same aggregate isolation protocol. This analysis revealed no significant difference in the amount of H3 in the aggregate fraction when comparing lysates derived from activated eggs that to do or do not contain NASP (Fig. 5, F and G; and Fig. S5 B). Thus, we conclude that upon egg activation, protein aggregates that contain H3 are preferentially targeted for degradation.

Taken together, we conclude that NASP does not directly protect H3 from degradation. Rather, NASP prevents H3 from aggregation. In the context of oogenesis and embryogenesis, aggregation and degradation are developmentally separated,

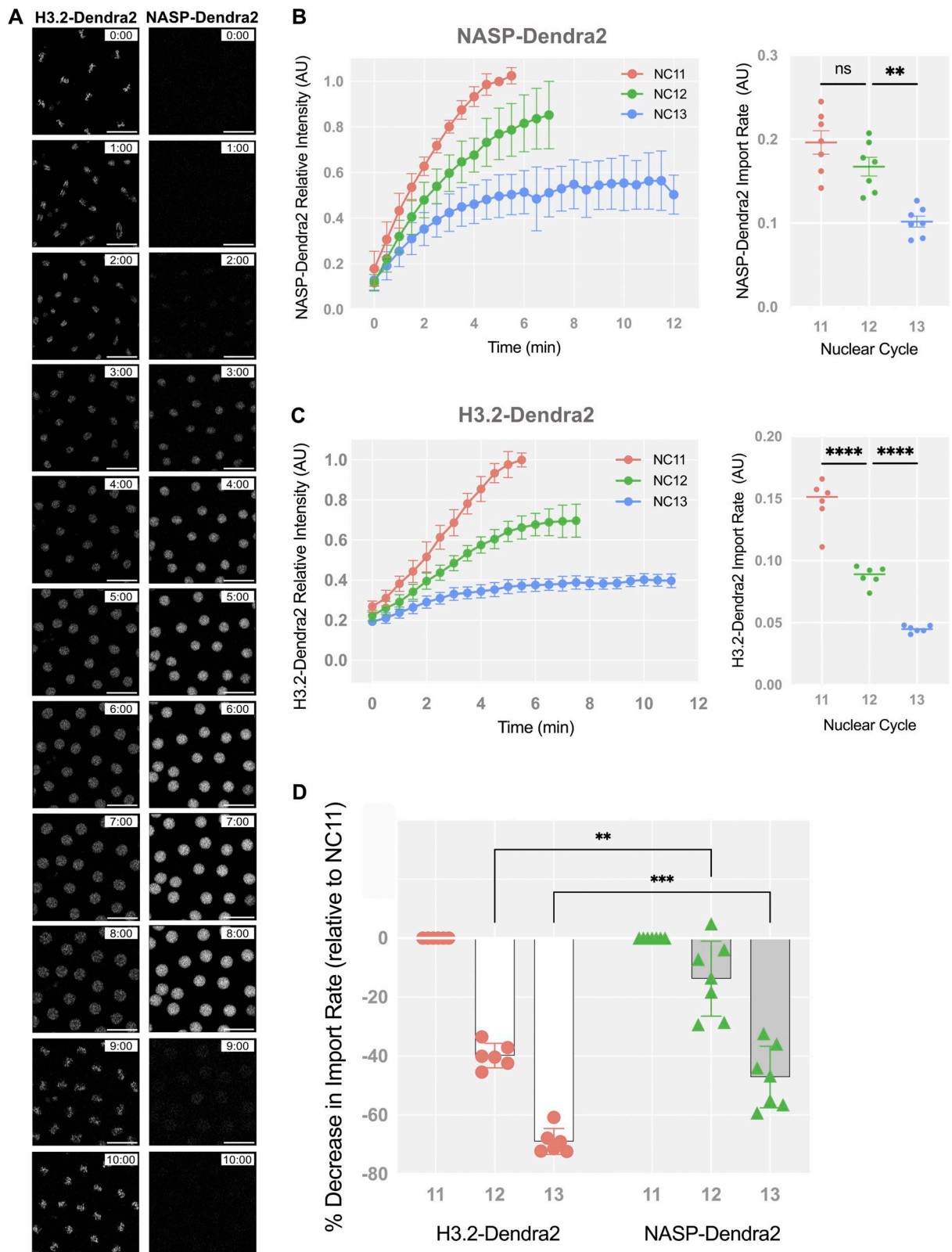

Figure 3.  **H3 and NASP have different nuclear import kinetics. (A)** Time-lapse of H3.2-Dendra2 (left) and NASP-Dendra2 (right) from metaphase 12 to metaphase 13. Scale bar represents 25 μm. **(B)** Relative intensity curves of NASP-Dendra2 for nuclear cycles NC 11–13 in WT embryos. *N* = 7 biological replicates. Data represent the mean ± standard deviation (SD). Statistical difference between the groups were analyzed by employing a one-way ANOVA Tukey's multiple comparison test. **(C)** Relative intensity curves of H3.2-Dendra2 for nuclear cycles NC 11–13 in WT embryos (same as in Fig. S1 A). *N* = 6 biological replicates. Data represent the mean ± standard deviation (SD). Statistical difference between the groups was analyzed by employing a one-way ANOVA Tukey's multiple comparison test. **(D)** The percent decrease in the nuclear import rate of H3.2-Dendra2 and NASP-Dendra2 for NC 11–13. *N* > 6

biological replicates. Data represent the mean ± standard deviation (SD). Statistical difference between the groups was analyzed by employing a two-way ANOVA with Geisser–Greenhouse correction and Fisher's least significant difference (LSD) to identify pairwise differences. In all figure panels, ns represents nonsignificant difference, while **P < 0.01, ***P < 0.001 and ****P < 0.0001.

allowing the two processes to be distinguished. In the absence of NASP, aggregation first occurs during oogenesis when H3 is being deposited into the egg chamber. Only upon egg activation are H3 protein aggregates targeted for degradation.

## Discussion

NASP is a H3-specific chaperone and the interaction between NASP and H3 has been well characterized on a biochemical and structural level (Campos et al., 2015; Zhang et al., 2018; Nabeel-Shah et al., 2014; Bowman et al., 2016; Liu et al., 2021; Tagami et al., 2004). How NASP functions in vivo to control H3 dynamics, however, is less well understood. By utilizing the early *Drosophila* embryo as a model system to study H3 dynamics, we found that NASP does not directly affect the nuclear import or export rates of histone H3.2 or H3.3. Rather, in the absence of NASP H3 protein levels are reduced, which leads to an indirect reduction in H3 nuclear import rate, less soluble H3 in the nucleoplasm and reduced H3 in chromatin. While NASP does localize to the nucleus in *Drosophila* embryos, NASP plays a critical role in the cytoplasm to chaperone H3. We found that H3 aggregation and degradation are developmentally uncoupled during *Drosophila* oogenesis and embryogenesis. Using this powerful developmental system, we have provided evidence that the key function of NASP in vivo is to prevent H3 aggregation rather than directly preventing H3 degradation.

To sustain the rapid nuclear cycles of early embryogenesis, embryos must be stockpiled with excess histones (Horard and Loppin, 2015). This provides a unique opportunity to understand how H3 nuclear import kinetics are affected by NASP. To our surprise, NASP does not directly affect the nuclear import kinetics of H3. Likewise, NASP does not act as a receptor to sequester H3 in the nucleus and prevent H3 from export. NASP directly binds to H3, and a number of NASP:H3 complexes have been identified from cell extracts (Campos et al., 2015; Tagami et al., 2004; Pardal and Bowman, 2022). H3 binding to importin-4 or importin-5, however, is mutually exclusive with NASP binding to H3 (Campos et al., 2015; Pardal and Bowman, 2022). Therefore, it is not clear how NASP could directly facilitate nuclear import of H3 if NASP is not part of the protein complex that is directly imported through the nuclear pore. NASP is localized to the nucleus in somatic cells, and the nuclear-specific function of NASP has yet to be defined. Given that H3 is not exported from the nuclei in *Drosophila* embryos (Shindo and Amodeo, 2019), it was possible that by binding to H3, NASP prevented nuclear export of H3 (Apta-Smith et al., 2018). Our

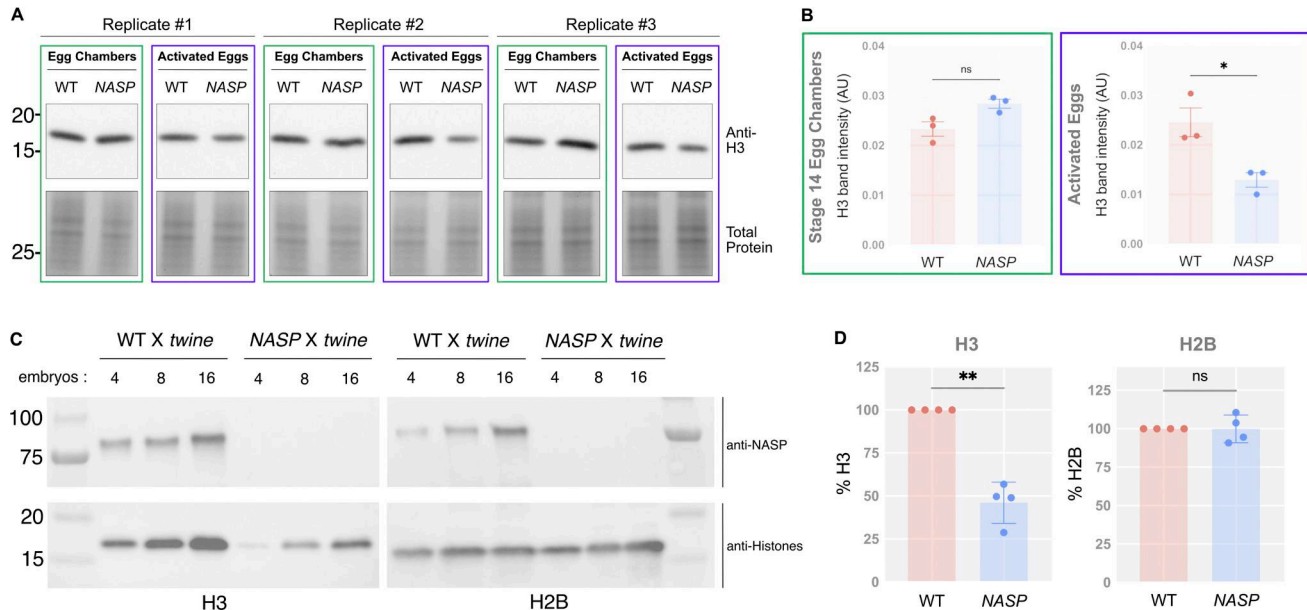

Figure 4. **H3m degradation happens only upon egg activation. (A)** Western blot analysis of stage 14 egg chambers and activated eggs collected from WT or *NASP*-mutant females. **(B)** Quantification of H3 band intensity in WT and *NASP* stage 14 egg chambers and activated eggs. H3 protein levels are normalized to the total protein level. N = 3 biological replicates. Data represent the mean ± standard deviation (SD). Statistical difference between the groups was analyzed by employing a paired *t* test. **(C)** Western blot analysis of activated eggs collected from WT or *NASP*-mutant females crossed to *twine*-mutant males. Embryos represent the number of individual embryos used for preparing the sample. **(D)** Quantification of H3 and H2B protein levels in activated eggs laid by WT and *NASP*-mutant females crossed to *twine*-mutant males. Proteins levels are normalized to the WT values. N = 3 biological replicates. Data represent the mean ± standard deviation (SD). Statistical difference between the groups was analyzed by employing a paired *t* test. In the figure panels, ns represents nonsignificant difference, while *P < 0.05 and **P < 0.01. Source data are available for this figure: SourceData F4.

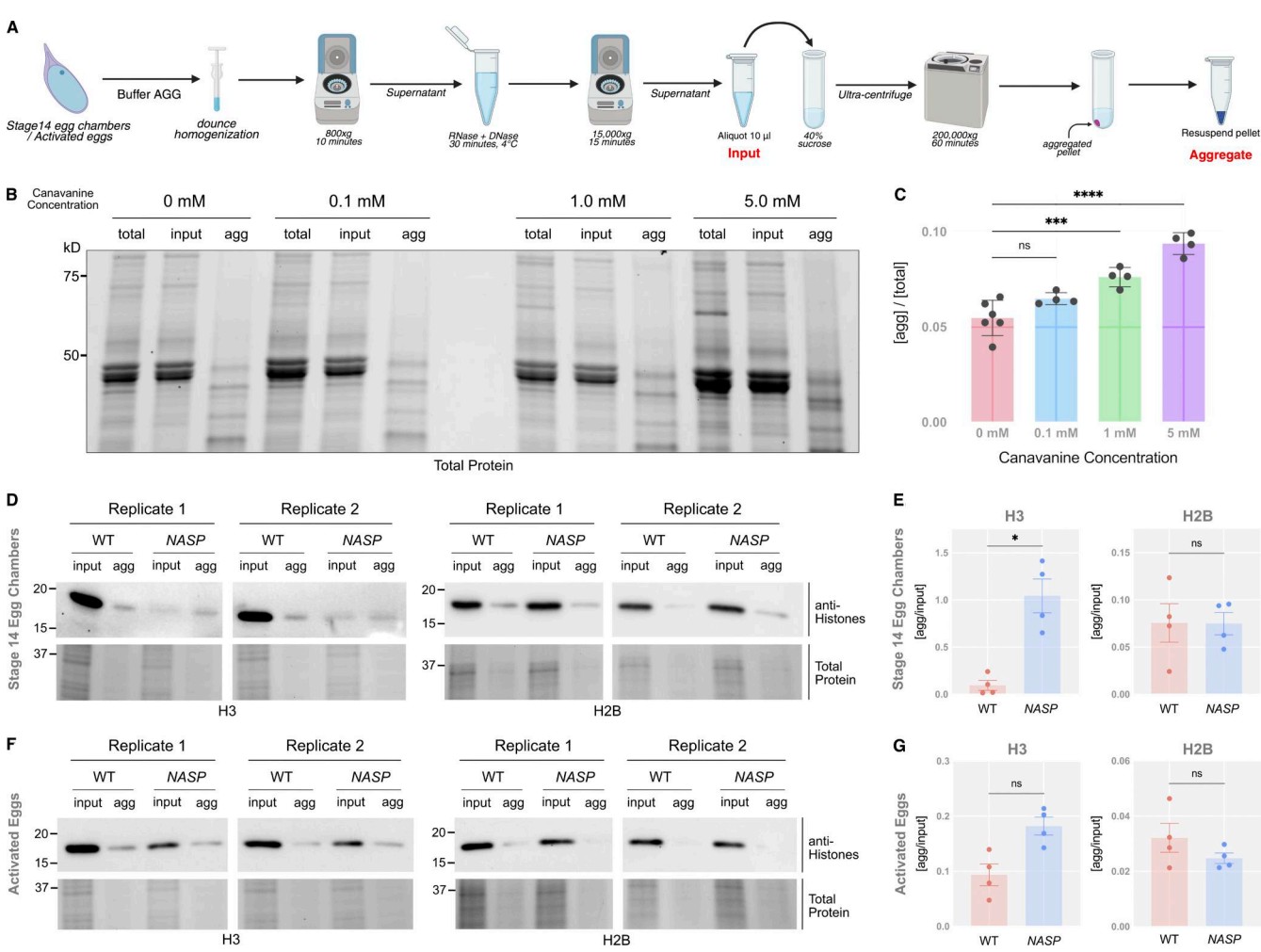

Figure 5. **NASP functions in the cytoplasm. (A)** Schematic of the aggregate isolation protocol used to separate aggregates from stage 14 egg chambers and early embryos. **(B)** SDS PAGE analysis for total and aggregated protein fractions in WT ovaries when treated with different concentrations of canavanine. **(C)** Ratio of protein in the aggregate fraction relative to the total protein in WT ovaries at different canavanine concentrations. $N > 4$ biological replicates. Data represent the mean ± standard deviation (SD). Statistical difference between the groups was analyzed by employing a one-way ANOVA Dunnett's multiple comparison test. **(D)** Western Blot analysis for input and aggregate fractions from stage 14 egg chambers from WT and *NASP*-mutant female flies. **(E)** Ratios of H3 and H2B in the aggregate fraction relative to input fractions in stage 14 egg chambers of WT and *NASP*-mutant female flies. $N = 4$ biological replicates. Data represent the mean ± standard deviation (SD). Statistical difference between the groups was analyzed by employing a paired *t* test. **(F)** Western blot analysis for input and aggregate fractions derived from activated eggs collected from WT or *NASP*-mutant females crossed to *twine*-mutant males. **(G)** Ratios of H3 and H2B in the aggregate fraction relative to input fractions from activated eggs collected from WT or *NASP*-mutant females crossed to *twine*-mutant males. $N = 4$ biological replicates. Data represent the mean ± standard deviation (SD). Statistical difference between the groups was analyzed by employing a paired *t* test. In the figure panels, ns represents nonsignificant difference, while $*P < 0.05$, $***P < 0.001$ and $****P < 0.0001$. Source data are available for this figure: SourceData F5.

results show that even in the absence of NASP, H3 is not exported from the nucleus. Thus, understanding the function of NASP in the nucleus remains enigmatic. In cancer cells, NASP functions in the nucleus to capture evicted nucleosomes upon PARP or topoisomerase inhibition (Moser et al., 2025). This function is specific to cancer cells; thus, the function of NASP in unperturbed human cells is still undefined.

In mammalian somatic cells, NASP prevents the degradation of H3, which occurs through CMA (Hormazabal et al., 2022). It is still unclear how NASP physically protects H3 from degradation. In this work, we demonstrated that H3 aggregation and H3 degradation are developmentally separated. Our work reveals that in the absence of NASP, H3 aggregation precedes

degradation. Thus, it appears that at least one critical function of NASP in vivo is to prevent H3 aggregation. While we think H3 aggregation likely proceeds degradation in somatic cells depleted of NASP, the limiting amounts of soluble histones make this difficult to address. By using activated eggs that are devoid of mitotic nuclei, we showed that NASP can function in the cytoplasm. In somatic cells and the early *Drosophila* embryo, however, NASP localizes to the nucleus. Because the cytoplasmic fraction of NASP is diffused in the *Drosophila* embryo, we cannot easily quantify the fraction of NASP in the nucleus during each nuclear cycle. Regardless, we propose that NASP functions both in the nucleus and the cytoplasm to prevent H3 aggregation. In the cytoplasm, protein aggregates are largely degraded through

autophagy (Mizushima, 2007; Mizushima et al., 2008; Lamark and Johansen, 2012). Consistent with this, CMA is the main H3 degradation pathway in somatic cells depleted of NASP (Hormazabal et al., 2022). Factors that drive autophagy are not present within the nucleus (Hale et al., 2013). Thus, NASP function in the nucleus could be extremely important to prevent potentially toxic nuclear aggregates of H3 that cannot be cleared through autophagy until nuclear envelope breakdown. Aggregates of H3 in the nucleus are likely to have similar properties to chromatin and could be more challenging to biochemically isolate and characterize. The pathway that degrades maternally inherited H3 aggregates in *Drosophila* has yet to be defined, and *Drosophila* lacks LAMP-2A, a critical factor for CMA (Kaushik and Cuervo, 2018).

Only about ~30% of embryos laid by *NASP*-mutant mothers hatch, and we have yet to define the molecular mechanism(s) that prevent these embryos from proper development (Tirgar et al., 2023). While embryos devoid of NASP have ~50% of the amount of H3 as embryos laid by WT embryos, we do not believe histone deficiency underlies the problems during embryogenesis. First, nearly half of the embryos laid by *NASP*-mutant mothers are arrested with one or two nuclei (Tirgar et al., 2023). Depleting half of the histone H3 pool in the early embryo should still provide enough histone to progress until nuclear cycle 13 (Shindo and Amodeo, 2019). Second, embryos laid by Jabba-mutant mothers (the H2A/H2B-specific chaperone) have vastly depleted maternal pools of H2A and H2B and yet proceed through embryogenesis relatively normally (Li et al., 2012). Third, embryos actively translate maternally deposited histone RNA, and this can compensate for reduced histone pools (Pérez-Roldán et al., 2025). One intriguing possibility is that the maternally transferred H3 protein aggregates cause toxicity in the early embryo. Protein aggregates are known to be toxic in several contexts (Soto and Pritzkow, 2018; Dubnikov et al., 2017; Wen et al., 2023; Stroo et al., 2017). Furthermore, aggregates containing high levels of histones could mimic chromatin and recruit histone chaperones (e.g., HIRA) or other histone-modifying enzymes into the aggregate, where they would be degraded or sequestered. Recent work in the mouse embryo has revealed that protein aggregates in the mouse oocyte can inhibit embryo survival if not degraded (Zaffagnini et al., 2024).

Embryos laid by *NASP*-deficient females have significantly reduced H3 levels that indirectly affect the import rate and nucleoplasmic supply of H3. Remarkably, NASP-deficient embryos also have a ~50% reduction in chromatin-associated H3. Although the embryos laid by NASP-deficient mothers have less H3 in their chromatin, the nuclei divide relatively normally, albeit a little slowly. The reduced incorporation of H3 into chromatin in these embryos presents an interesting question about genome packaging in these embryos. For example, are nucleosomes randomly distributed, or are there hotspots where nucleosomes must be positioned? How does a reduced number of nucleosomes affect processes like zygotic transcription and DNA replication? Using a tractable developmental model system will allow us to address these questions while discerning the function of NASP during embryogenesis and beyond.

# Materials and methods

## Strain list and stock generation

WT—Oregon R (OrR)

*NASP* null mutant (*NASP*)—*w[1118]; Df(3R)Exel6150, P{w[+mC] = XP-U} Exel6150/TM6B, Tb[1]/NASP[2]*

Df(3R)—*w[1118]; Df(3R)Exel6150, P{w[+mC] = XP-U} Exel6150/ TM6B, Tb[1]*

NASP-Dendra2—*y[1] M{vas-Cas9}ZH2A w[1118]; NASP-Dendra2*

H3.2-Dendra2—*y,w; 1xHisC.H3-Dendra2*

H3.3-Dendra2—*y,w; H3.3A-Dendra2/CyO*

twine—*twe[1] cn[1] bw[1]/CyO* and *y[1] w[67c23]; P{w[+mC] = lacW} twe[k08310]/CyO.*

To generate a Dendra2-tagged allele of *NASP*, a single gRNA targeting the C-terminal of *NASP* was cloned into the pU6-BbsI plasmid as described (Gratz et al., 2015). The gRNA was identified using the DRSC Find CRISPRs tool (http://www.flyrnai.org/crispr2/index.html). The NASP gRNA target site was cloned into the pU6-BbsI plasmid using forward primer 5′-CTTCGGCAC CGACTCGGTGCCACT-3′ and reverse primer 5′-AAACAGTGG CACCGAGTCGGTGC-3′. The recovery vector was assembled by Gibson Assembly using the primers: Dendra2_forward 5′- GGC GGCTCAGGGGGTAGTATGAACACCCCGGGAA-3′, Dendra2_reverse 5′- GATTATCTTTAACGTACGTTACCACACCTGGCTGGG-3′, NASP_forward 5′- CCCAGCCAGGTGTGGTAACGTACGTTA AAGATAATC-3′, and NASP_reverse 5′- TTCCCGGGGGTGTTCA TACTACCCCCTGAGCCGCC-3′. The gRNA-expressing plasmid and the NASP-Dendra2 recovery vector were injected into a *vas-Cas9* expression stock (Best Gene Inc.) and balanced over *TM3*. The flies were self-crossed and screened for lack of the *TM3* phenotype to establish a homozygous stock.

## Embryo collection

*NASP/Df(3R)* or OrR female flies were yeast fed for 4 days at room temperature. The ovaries were dissected in 1X PBS, and stage 14 egg chambers were isolated.

For embryo collection, the fly strains or crosses were set up at room temperature in bottles capped by a grape juice agar plate with some wet yeast. For western blotting, 0–2-h (after egg laying—AEL) embryos were collected. To obtain activated eggs from *twine*-mutant crosses, 0–1-h (AEL) embryos were collected. To image activated eggs, overnight embryos (16–24 h AEL) were collected. All the embryo collection samples were dechorionated with 50% bleach for up to 2 min and thoroughly washed with water twice. The dechorionated embryos were either flash-frozen with liquid nitrogen and stored at –80°C or processed for further experiments.

## Cell culture

*Drosophila melanogaster* S2 cell stocks were maintained in T75 flasks at room temperature in Schneider's *Drosophila* Medium (1X) (Cat#21720-024; Gibco) supplemented with 10% Fetal Bovine Serum (FBS) (Cat#100–108; Gemini-Bio), 100 U/ml Penicillin, and 100 μg/ml Streptomycin (Cat#15140-122; Gibco). Cells were passaged 1:5 dilution when the flasks reached >90% confluency. Cells were counted and plated in 6-well plates 24 h prior to drug treatment.

## Live imaging

### Images for nuclear import and chromatin analysis

For live imaging, *Drosophila* embryos were collected from fresh grape juice plates after allowing flies to lay for 2 h. Embryos were then dechorionated with 30% bleach solution for 1 min and 30 s and washed with water. Embryos were mounted on a glass-bottom dish in deionized water, and images were acquired using a Zeiss LSM 980 confocal microscope with Airyscan-2 the 40×, 1.2 NA water objective. Embryos laid by *H3.2-Dendra2, H3.2-Dendra2; NASP; NASP-GFP*, and *NASP-Dendra2* mothers were imaged with the 488 nm laser at 2% power using the CO8-Y Airyscan Setting. *H3.3-Dendra2* embryos were imaged at 0.5% laser power. All z-stacks comprised 16 planes spaced 1 µm apart. The stage was heated at 25°C.

For nuclear import and chromatin analysis, all were imaged every 45 s for 2 h, with a 712 × 712-pixel resolution, at 3× zoom, and a 35.88 ms frame time. The pixel size of all images was 0.099 × 0.099 µs.

### Images for nuclear export

For nuclear export, imaging was performed as described for nuclear import with the following exceptions. Nuclear export imaging was done with interactive bleaching mode on ZEN Blue Software. The Dendra2 tag was photoconverted from green to red using 60 iterations of the 405 nm laser at 3% power with an exposure speed of 1.37 µs/pixel. A 4 µm diameter circle stencil was used to outline the nuclei meant to be photoconverted. The nucleus was converted in the middle of the nuclear cycle and imaged for 15-s intervals until the end of the nuclear cycle. Images were captured at a 576 × 576 pixel resolution, at 4 × zoom, with a 66.55-ms frame time, and a pixel size of 0.092 µm × 0.092 µm.

### Photobleaching

To determine potential photobleaching during image acquisition, two embryos of the same age were imaged from the interphase nucleus and metaphase chromatin for NC10. Then a subregion of one embryo was imaged with the experimental import settings until NC13. Once the imaged embryo reached NC13, both were imaged at the interphase nucleus and metaphase chromatin of NC13. The continuously imaged area was compared with the outside area and the unimaged embryo. These comparisons showed minimal photobleaching, and therefore, no numerical photobleaching corrections were applied to our data.

### Segmentation and image analysis

Images were processed using ZEN 3.3 (blue edition) and 3D Airyscan processed with a strength of 3.7. Scenes were separated and converted to individual TIFF files.

The z-stack for the time point of metaphase chromatin for each nuclear cycle was sum projected in FIJI. The chromatin and cytoplasm of these images were then segmented using the pixel classification + object classification features on Ilastik. A CSV file was exported containing the total intensity.

To obtain nuclear import curves, live images were divided into individual cycles and segmented using pixel classification +

object classification on Ilastik. CSV files contained total intensity values, which were subtracted by the nuclear intensity of the first frame of each cycle to plot the total intensities starting at 0. To obtain relative intensities, the total intensity values were normalized to the end of nuclear cycle 11 for each embryo. Cell cycle duration was measured from metaphase to metaphase of the next cycle.

### Statistical analysis

To calculate import rates, the initial slope of the import curves was calculated using a simple linear regression. Two-way ANOVA tests were performed to evaluate the statistical significance between different genotypes.

### Western blotting

Protein lysates were prepared by homogenizing dechorionated embryos in 1.5-ml Eppendorf tube using a pestle in 2X Laemmli Sample Buffer (Cat#1610737; Bio-Rad) supplemented with 50 mM DTT. Lysates were boiled for 10 min and loaded on a 4–15% Mini-PROTEAN TGX Stain-Free Gel (Cat#4568086; Bio-Rad). After electrophoresis, gels were activated and imaged using a Bio-Rad ChemiDoc MP Imaging System using default manufacturer parameters. Protein was transferred to Immobilon-P PVDF Transfer Membrane (Ref#IPVH00010; Millipore) using the Bio-Rad Trans-Blot Turbo Transfer System, and membranes were imaged using the Bio-Rad ChemiDoc MP Imaging System using default manufacturer parameters. The transfer membrane was blocked with 5% nonfat milk in 1X TBS-T (140 mM NaCl, 2.5 mM KCl, 50 mM Tris HCl, pH 7.4, and 0.1% Tween-20) for 10 min. Blots were incubated with primary antibodies—anti-NASP—1:2,000 (lab-generated), anti-H3-HRP—1:1,000 (Cat#ab21054; Abcam), anti-H2B—1:1,000 (Cat#mAb52484; Abcam)—for 1 h at room temperature, washed, and incubated with secondary antibodies—HRP anti-mouse—1: 20,000 (Cat#715-035-150; Jackson ImmunoResearch), HRP anti-rabbit—1:25,000 (Cat#111-035-003; Jackson ImmunoResearch) for 30 min at room temperature. Blots were then washed with TBS-T and incubated with Clarity ECL Western Solution (Cat#1705061; Bio-Rad) for 4 min prior to imaging. Blots were imaged using the Bio-Rad ChemiDoc MP Imaging System. The western blots were quantified using Bio-Rad Image Lab software.

### Aggregate isolation

A standard aggregate isolation protocol was adapted from Chen et al. (2024) to isolate protein aggregates from *Drosophila* egg chambers and activated eggs. Stage 14 egg chambers or activated eggs were homogenized using a dounce homogenizer with a B-type pestle in 100 µl of buffer AGG (30 mM Tris-Cl, pH 7.5, 1 mM DTT, 40 mM NaCl, 3 mM $CaCl_2$, 3 mM $MgCl_2$, 5% glycerol, and 1% Triton X-100 supplemented with Roche cOmplete EDTA-free Protease Inhibitor Cocktail, Cat#04693132001). The homogenized lysate was transferred to a 1.5-ml Eppendorf tube and centrifuged at 800×*g* for 10 min at 4°C to remove cell debris. The supernatant was transferred to a new 1.5-ml Eppendorf tube and treated with 100 µg/ml RNase A (Cat#EN0531; Thermo Fisher Scientific) and 100 µg/ml DNase I (Cat#M030S; NEB) for 30 min

on ice. Nuclease-treated samples were centrifuged at 10,000×*g* for 15 min at 4°C (same Beckman Coulter FA-241.5 rotor). The resulting supernatant is the "input lysate" fraction, and an aliquot of this fraction was saved for western blotting. The remaining lysate was loaded on top of 1 ml of a 40% sucrose pad, and an additional 750 µl of Buffer AGG was added to the lysate in the ultracentrifugation tube (Cat#347357; Beckman Coulter Polypropylene Centrifuge Tubes). Samples were subjected to ultracentrifugation for 1 h at 200,000×*g* at 4°C (Optima TL Ultracentrifuge using a Beckman TLS-55 rotor). Most of the supernatant was removed, leaving 10–15 µl of sample at the bottom of the ultracentrifuge tube. An additional 20 µl of Buffer AGG was added to rigorously resuspend the remaining fraction, which constitutes the "aggregate" fraction.

For the positive control experiments, ovaries were extracted from yeast fed OrR female flies in 1X PBS. The ovaries were treated with different concentrations of canavanine (Cat#C9758; Sigma-Aldrich)—0.0, 0.1, 1.0, and 5.0 mM and azetidine-2-carboxylic acid (Cat#A0760; Sigma-Aldrich)—0.0, 5.0, and 10.0 mM for 3 h at room temperature. The ovaries were then lysed via dounce homogenization, and the lysate was used for the aggregate isolation assay.

For the second positive control, *Drosophila* S2 cells were plated at $9 \times 10^6$ cells/well and treated with different concentrations of canavanine (Cat#C9758; Sigma-Aldrich)—0.0, 0.1, 0.5, and 1.0 mM for 3 h, and the cells were then harvested and lysed in RIPA Buffer (140 mM NaCl, 1% NP40, 1 mM EDTA, 0.1% Na-deoxycholate, and 0.1% SDS, supplemented with Roche cOmplete EDTA-free Protease Inhibitor Cocktail, Cat#04693132001). The lysate was then utilized for the aggregate isolation assay.

### Online supplemental material

Fig. S1 shows the normalized nuclear import rates and total intensity curves for nuclear export analysis of H3.2-Dendra2 and H3.3-Dendra2 in WT and *NASP*-mutant embryos. Fig. S2 shows the validation of the NASP-Dendra2 fly line and the H3.2-Dendra2 nuclear import rates excluding the chromatin-bound H3.2 intensity. Fig. S3 shows that the embryos laid by WT females crossed to *twine*-mutant males fail to develop with reduction in total H3 levels but not H2B levels. Fig. S4 shows the validation of the aggregate isolation assay and the H3 levels in different fractions of the aggregate isolation assay. Fig. S5 shows additional replicates of the aggregate isolation assay in stage 14 egg chambers and activated eggs used for quantification. Video 1 shows live imaging of H3.2-Dendra2 through NC10-14. Video 2 shows live-imaging of NASP-Dendra2 through NC10-14.

## Data availability

All raw and processed images are available upon reasonable request.

## Acknowledgments

Essential fly stocks were provided by the Bloomington *Drosophila* Stock Center (NIHP40OD018537). This research was supported by National Institutes of Health (NIH) grants 2R35GM128650 (to J.T. Nordman) and R35GM150853 to A.A. Amodeo. Open Access funding provided by Vanderbilt University.

Author contributions: Mohit Das: conceptualization, formal analysis, investigation, and writing—original draft, review, and editing. Eli Coronado-Chavez: data curation, formal analysis, investigation, methodology, visualization, and writing—original draft. Anusha D. Bhatt: formal analysis, methodology, and writing—review and editing. Reyhaneh Tirgar: methodology. Amanda A. Amodeo: methodology, supervision, and writing—review and editing. Jared T. Nordman: conceptualization, funding acquisition, supervision, and writing—original draft, review, and editing.

Disclosures: The authors declare no competing interests exist.

Submitted: 5 December 2025

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

**Supplemental material**

Figure S1.  **Normalized nuclear import rates for H3.2 and H3.3 Dendra. (A)** Normalized relative intensity curves of H3.2-Dendra2 for nuclear cycles (NC) 11–13 in WT embryos. Values normalized to maximum H3.2-Dendra2 intensity in NC11. **(B)** Normalized relative intensity curves of H3.2-Dendra2 for nuclear cycles (NC) 11–13 in NASP-deficient embryos. Values normalized to maximum H3.2-Dendra2 intensity in NC11. **(C)** Normalized nuclear import rates of H3.2-Dendra2 in WT (blue) and NASP-deficient (red) embryos. Simulated import rate for H3.2 at 50% WT levels (black). $N = 6$ biological replicates. Data represent the mean ± standard deviation (SD). Statistical difference between groups were analyzed by employing a two-way ANOVA Tukey's multiple comparison test between the WT and *NASP* data points. **(D)** Normalized relative intensity curves of H3.3-Dendra2 for nuclear cycles (NC) 11–13 in WT embryos. Values normalized to maximum H3.2-Dendra2 intensity in NC11. **(E)** Normalized relative intensity curves of H3.3-Dendra2 for nuclear cycles (NC) 11–13 in NASP-deficient embryos. Values normalized to maximum H3.2-Dendra2 intensity in NC11. **(F)** Normalized nuclear import rates of H3.3-Dendra2 in WT (blue) and NASP-deficient (red) embryos. $N = 6$ biological replicates. Data represent the mean ± standard deviation (SD). Statistical difference between groups were analyzed by employing a two-way ANOVA Tukey's multiple comparison test between the WT and *NASP* data points. **(G)** Normalized relative intensity curves for simulation of H3 nuclear import at 50% initial concentration. Values normalized to maximum H3.2-Dendra2 intensity in NC11. **(H)** Slope for nuclear import rate calculation for simulation of H3 nuclear import at 50% initial concentration. **(I)** Cell cycle duration for NC11-13 in WT (blue) and NASP-deficient (red) embryos. Simulated cell cycle duration for NC13 at 50% WT H3.2 levels (black). $N = 6$ biological replicates. Data represent the mean ± standard deviation (SD). Statistical difference between groups were analyzed by employing a two-way ANOVA Tukey's multiple comparison test between the WT and *NASP* data points. In the figure panels, ns represents nonsignificant difference, while *P < 0.05, **P < 0.01, ***P < 0.001 and ****P < 0.0001. **(J)** Un-normalized nuclear export measurement for H3.2-Dendra2 in WT and NASP-deficient embryos in NC12. Time point 0 represents Nuclear Envelope Breakdown (NEB). $N = 3$ biological replicates. Data represents mean ± standard deviation (SD). **(K)** Un-normalized nuclear export measurement for H3.3-Dendra2 in WT and NASP-deficient embryos in NC12. Time point 0 represnts Nuclear Envelope Breakdown (NEB). $N = 3$ biological replicates. Data represents the mean ± standard deviation (SD).

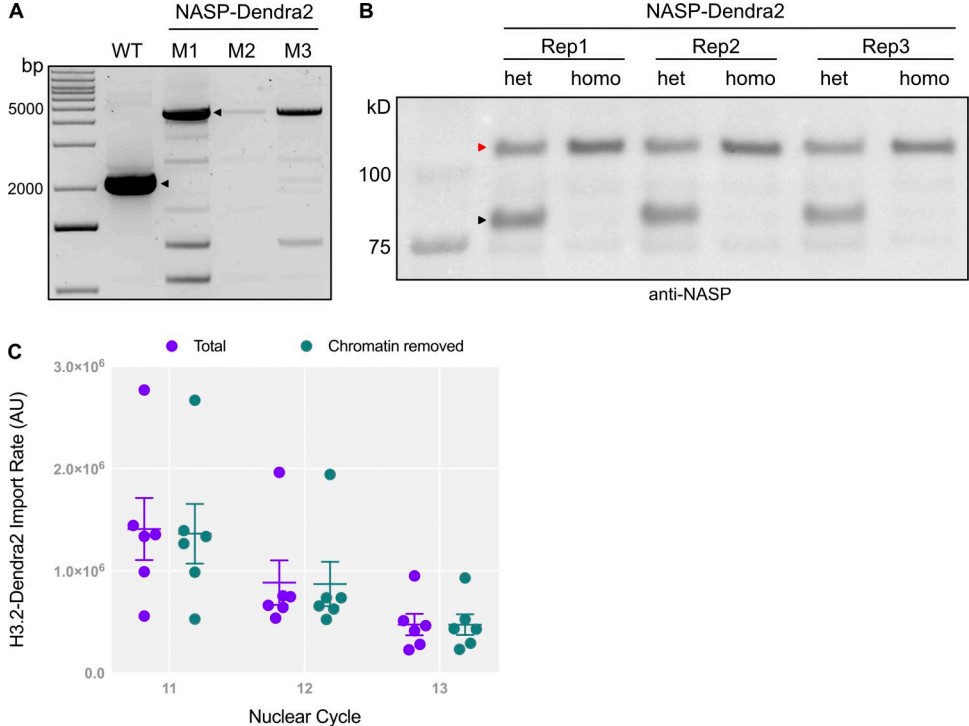

Figure S2. **Validation of NASP-Dendra fly line. (A)** PCR analysis of the NASP endogenous locus shows ~3 kb shift in the NASP-Dendra–tagged flies as compared with the WT flies. M1—male #1, M2—male #2, and M3—male #3. **(B)** Western blot analysis of ovary extracts from heterozygous (het) and homozygous (homo) *NASP-Dendra* female flies confirm the expression of NASP-Dendra (red arrowhead represents Dendra-tagged NASP, and the black arrowhead represents WT NASP). Three biological replicates are labelled as Rep1, Rep2, and Rep3. **(C)** H3.2 Dendra nuclear import rates are calculated from unnormalized total nuclear intensities (purple) and non-chromatin–associated intensities (teal). Source data are available for this figure: SourceData FS2.

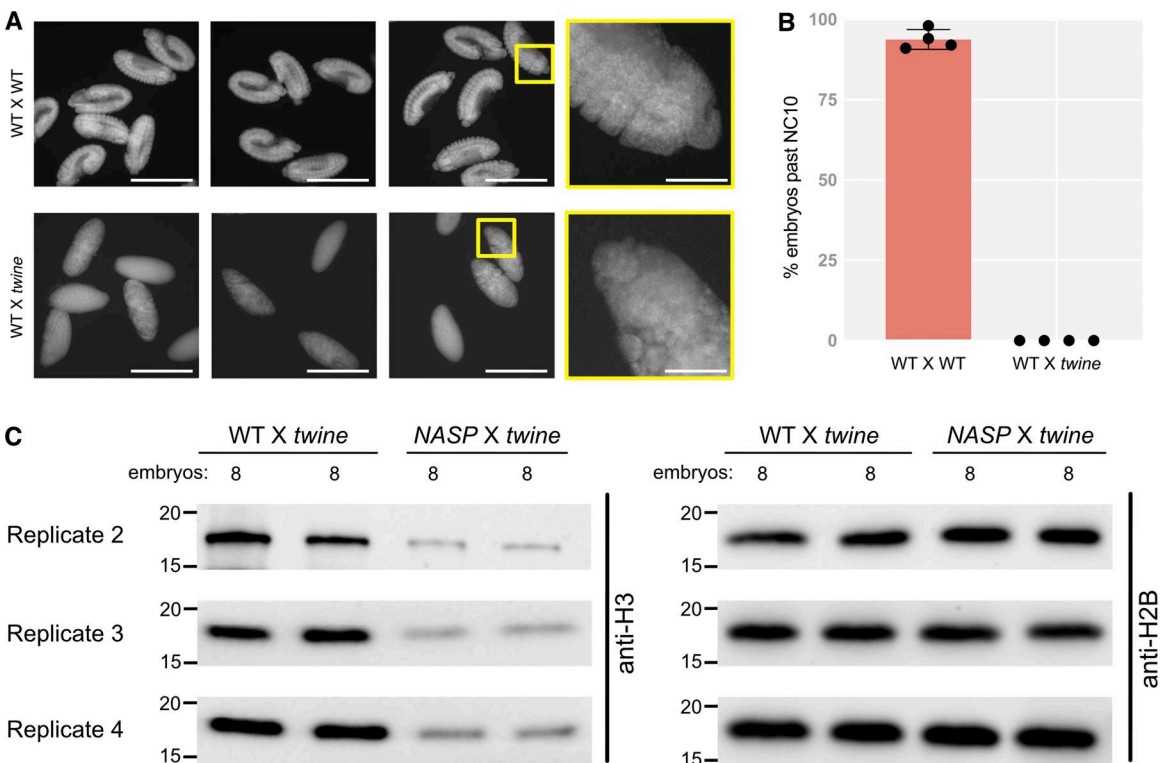

Figure S3.   **Embryos laid by WT females crossed to *twine*-mutant males fail to develop. (A)** Representative images of DAPI-stained eggs from WT females crossed to WT males (WT X WT) and WT females crossed to *twine*-mutant males (WT X *twine*). Imaging was performed at 10X or 60X (yellow boxes). Scale bars represent 500 and 83 μm (inset). **(B)** Percentage of eggs laid that progress past nuclear cycle 10 from WT X WT and WT X *twine*-mutant crosses. **(C)** Western blot analysis of activated eggs collected from WT or *NASP*-mutant females crossed to *twine*-mutant males used in quantification in Fig. 4 B (replicates 3 and 4). Embryos indicate the number of embryos used for the western blots. Source data are available for this figure: SourceData FS3.

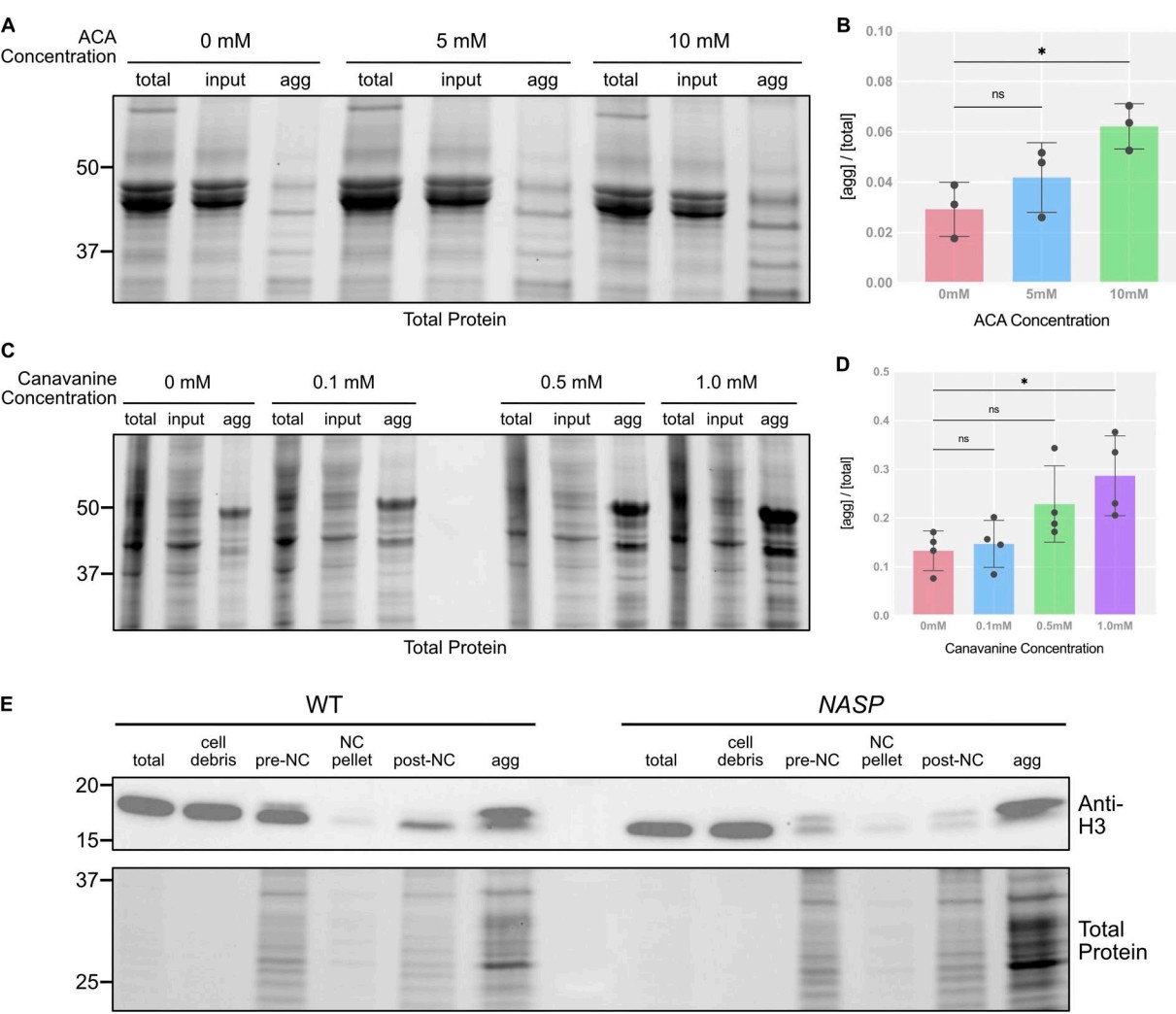

Figure S4. **Validation of aggregate isolation assay. (A)** SDS-PAGE analysis for total, input, and aggregate fractions from ovaries collected from WT female flies and treated with different concentrations of azetidine-2-carboxylic acid. **(B)** Ratios of the aggregated fractions relative to the total protein at different azetidine-2-carboxylic acid concentrations quantified from the SDS-PAGE. $N$ = 3 biological replicates. Data represent the mean ± standard deviation (SD). Statistical difference between groups was analyzed by employing a one-way ANOVA Dunnett's multiple comparisons test. **(C)** SDS-PAGE analysis for total, input, and aggregate fractions from S2 cells treated with different concentrations of canavanine. **(D)** Ratios of the aggregated fractions relative to the total protein at different canavanine concentrations quantified from the SDS PAGE. $N$ = 4 biological replicates. Data represent the mean ± standard deviation (SD). Statistical difference between groups was analyzed by employing a one-way ANOVA Dunnett's multiple comparisons test. **(E)** Western blot for H3 levels at different fractions of the aggregate isolation assay in WT and $NASP$ stage 14 egg chambers. The total and cell debris fractions are diluted 1:10 to prevent saturation of the chemiluminescent signal. In the figure panels, ns represents a nonsignificant difference, while *P < 0.05. Source data are available for this figure: SourceData FS4.

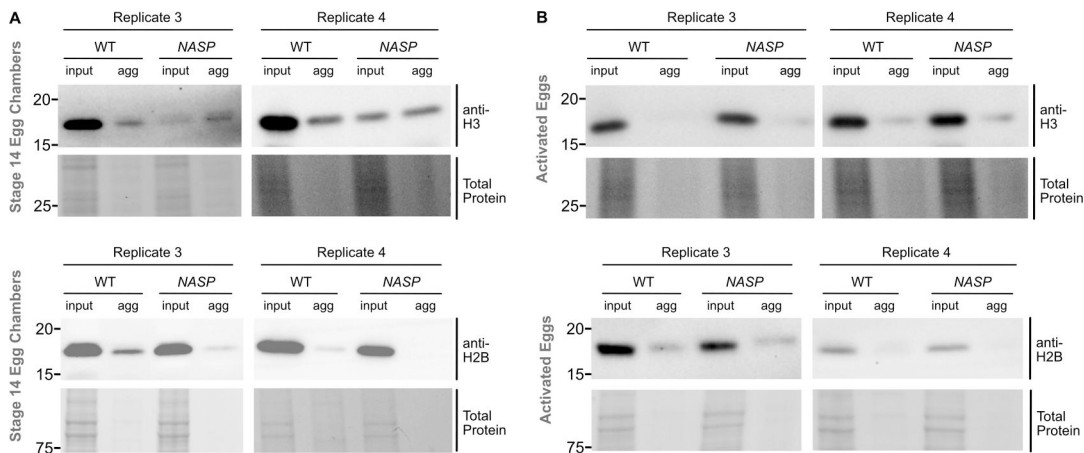

Figure S5.   **Western blots of histone aggregation across multiple replicates. (A)** Western blot analysis for input and aggregate fractions from stage 14 egg chambers from WT and *NASP* mutant female flies used in quantification in Fig. 4 E (replicates 3 and 4). **(B)** Western blot analysis for input and aggregate fractions derived from activated eggs collected from WT or *NASP*-mutant females crossed to *twine*-mutant males used in quantification in Fig. 4 G. (Replicates 3 and 4). Source data are available for this figure: SourceData FS5.

Video 1.   **Live imaging of H3.2-Dendra2 through NC10–14.**

Video 2.   **Live imaging of NASP-Dendra2 through NC10–14.**

