## [Peer Review File · The Journal of Cell Biology]

NASP functions in the cytoplasm to prevent histone H3 aggregation during early embryogenesis

Mohit Das, Eli Coronado-Chavez, Anusha Bhatt, Reyhaneh Tirgar, Amanda Amodeo, and Jared Nordman

Corresponding Author(s): Jared Nordman, Vanderbilt University

Review Timeline:

Submission Date:	2025-12-05
Editorial Decision:	2025-12-16
Revision Received:	2026-02-26
Editorial Decision:	2026-03-24
Revision Received:	2026-03-27

Monitoring Editor: Hiroshi Kimura

Scientific Editor: Gabriele Stephan

Transaction Report:

DOI: <https://doi.org/10.1083/jcb.202511182>

Revision 0

Review #1

1. Evidence, reproducibility and clarity:

Evidence, reproducibility and clarity (Required)

****Summary:****

The authors investigate the function of the H3 chaperone NASP, which is known to bind directly to H3 and prevent degradation of soluble H3. What is unclear is where NASP functions in the cell (nucleus or cytoplasm), how NASP protects H3 from degradation (direct or indirect), and if NASP affects H3 dynamics (nuclear import or export). They use the powerful model system of *Drosophila* embryos because the soluble H3 pool is high due to maternal deposition and they make use of photoconvertible Dendra-tagged proteins, since these are maternally deposited and can be used to measure nuclear import/export rates.

Using these systems and tools, they conclude that NASP affects nuclear import, but only indirectly, because embryos from NASP mutant mothers start out with 50% of the maternally deposited H3. Because of the depleted H3 and reduced import rates, NASP deficient embryos also have reduced nucleoplasmic and chromatin-associated H3. Using a new Dendra-tagged NASP allele, the authors show that NASP and H3 have different nuclear import rates, indicating that NASP is not a chaperone that shuttles H3 into the nucleus. They test H3 levels in embryos that have no nuclei and conclude that NASP functions in the cytoplasm, and through protein aggregation assays they conclude that NASP prevents H3 aggregation.

****Major comments:****

The text was easy to read and logical. The data are well presented, methods are complete, and statistics are robust. The conclusions are largely reasonable. However, I am having trouble connecting the conclusions in text to the data presented in Figure 4.

First, I'm confused why the conclusion from Figure 4A is that NASP functions in the cytoplasm of the egg. Couldn't NASP be required in the ovary (in, say, nurse cell nuclei) to stimulate H3 expression and deposition into the egg? The results in 4A would look the same if the mothers deposit 50% of the normal H3 into the egg. Why is NASP functioning specifically in the cytoplasm when it is also so clearly imported into the nucleus? Maybe

NASP functions wherever it is, and by preventing nuclear import, you force it to function in the cytoplasm. I do not have additional suggestions for experiments, but I think the authors need to be very clear about the different interpretations of these data and to discuss WHY they believe their conclusion is strongest.

Second, an alternate conclusion from Figure 4D/E is that mothers are depositing less H3 protein into the egg, but the same total amount is being aggregated. This amount of aggregated protein remains constant in activated eggs, but additional H3 translation leads to more total H3? The authors mention that additional translation can compensate for reduced histone pools (line 416).

As the function of NASP in the cytoplasm (when it clearly imports into the nucleus) and role in H3 aggregation are major conclusions of the work, the authors need to present alternative conclusions in the text or complete additional experiments to support the claims. Again, I do not have additional suggestions for experiments, but I think the authors need to be very clear about the different interpretations of these data and to discuss WHY they believe their conclusion is strongest.

Data presentation:

Overall, I suggest moving some of the supplemental figures to the main text, adding representative movie stills to show where the quantitative data originated, and moving the H3.3 data to the supplement. Not because it's not interesting, but because H3.3 and H3.2 are behaving the same.

Fig 1:

It would strengthen the figure to include representative still images that led to the quantitative data, mostly so readers understand how the data were collected.

The inclusion of a "simulated 50% H3" in panel C is confusing. Why?

I would also consider normalizing the data between A and B (and C and D) by dividing NASP/WT. This could be included in the supplement (OPTIONAL)

Fig S1:

The data simulation S1G should be moved to the main text, since it is the primary reason the authors reject the hypothesis that NASP influences H3 import rates.

Fig 2:

Once again, I think it would help to include a few representative images of the photoconverted Dendra2 in the main text.

I struggled with A/B, I think due to not knowing how the data were normalized. When I realized that the WT and NASP data are not normalized to each other, but that the NASP values are likely starting less than the WT values, it made way more sense. I suggest switching the order of data presentation so that C-F are presented first to establish that there is less chromatin-bound H3 in the first place, and then present A/B to show no change in nuclear export of the H3 that is present, allowing the conclusion of both less soluble AND chromatin-bound H3.

Fig S2:

If M1-M3 indicate males, why are the ovaries also derived from males? I think this is just confusing labeling.

Supplemental Movie S1:

Beautiful. Would help to add a time stamp (OPTIONAL).

Fig 3:

Panel C is the same as Fig S1A (not Fig 1A, as is said in the legend), though I appreciate the authors pointing it out in the legend. Also see line 276.

Panel D is a little confusing, because presumably the "% decrease in import rate" cannot be positive (Y axis). This could be displayed as a scatter (not bar) as in Panels B/C (right) where the top of the Y axis is set to 0.

Fig S3:

A: What do the different panels represent? I originally thought developmental time, but now I think just different representative images? Are these age-matched from time at egg lay?

C: What does "embryos" mean? Same question for Fig 4A.

Fig 4:

A: What does "embryos" mean? Number of embryos? Age in hours?

C: Not sure the workflow figure panel is necessary, as I can't tell what each step does. This is better explained in methods. However I appreciated the short explanation in the text (lines 314-5).

****Minor comments:****

The authors should describe the nature of the NASP alleles in the main text and present evidence of robust NASP depletion, potentially both in ovaries and in embryos. The antibody works well for westerns (Fig S2B). This is sort of demonstrated later in Figure 4A, but only in NAAP x twine activated eggs.

Lines 163, 251, 339: minor typos

Line 184: It would help to clarify- I'm assuming cytoplasmic concentration (or overall) rather than nuclear concentration. If nuclear, I'd expect the opposite relationship. This occurs again when discussing NASP (line 267). I suspect it's also not absolute concentration, but relative concentration difference between cytoplasm and nucleus. It would help clarify if the authors were more precise.

Line 189: Given that the "established integrative model" helps to reject the hypothesis that NASP is involved in H3 import, I think it's important to describe the model a little more, even though it's previously published.

Line 203: "The measured rate of H3.2 export from the nucleus is negligible" clarify this is in WT situations and not a conclusion from this study.

Line 201: How can the authors be so sure that the decrease in WT is due to "the loss of non-chromatin bound nucleoplasmid H3.2-Dendra2?"

Line 217: In the conclusion, the authors indicate that NASP indirectly affects soluble supply of H3 in the nucleoplasm. I do believe they've shown that the import rate effect is indirect, but I don't know why they conclude that the effect of NASP on the soluble nucleoplasmic H3 supply is indirect. Similarly, the conclusion is indirect on line 239. Yet, the authors have not shown it's not direct, just assumed since NASP results in 50% decrease to deposited maternal histones.

Line 292: What is the nature of the NASP "mutant?" Is it a null? Similarly, what kind of "mutant" is the twine allele? Line 295.

Line 316: Why did the authors use stage 14 egg chambers here when they previously used embryos? This becomes more clear later shortly, when the authors examine activated eggs, but it's confusing in text.

Lines 343-348: It's unclear if the authors are drawing extended conclusions here or if they are drawing from prior literature (if so, citations would be required). For example, why during oogenesis/embryogenesis are aggregation and degradation developmentally separated?

Lines 386-7: I do not understand why the authors conclude that H3 aggregation and degradation are "developmentally uncoupled" and why, in the absence of NASP, "H3 aggregation precedes degradation."

Line 395: Why suddenly propose that NASP also functions in the nucleus to prevent aggregation, when earlier the authors suggest it functions only in the cytoplasm?
Lines 409-413: The authors claim that histone deficiency likely does not cause the embryonic arrest seen in embryos from NASP mutant mothers. This is because H3 is reduced by 50% yet some embryos arrest long before they've depleted this supply. However, the authors also showed that H3 import rates are affected in these embryos due to lower H3 concentration. Since the early embryo cycles are so rapid, reduced H3 import rates could lead to early arrest, even though available H3 remains in the cytoplasm.

2. Significance:

Significance (Required)

The significance of the work is conceptual, as NASP is known to function in H3 availability but the precise mechanism is elusive. This work represents a necessary advance, especially to show that NASP does not affect H3 import rates, nor does it chaperone H3 into the nucleus. However, the authors acknowledge that many questions remain. Foremost, why is NASP imported into the nucleus and what is its role there?

I believe this work will be of interest to those who focus on early animal development, but NASP may also represent a tool, as the authors conclude in their discussion, to reduce histone levels during development and examine nucleosome positioning. This may be of interest to those who work on chromatin accessibility and zygotic genome activation.

I am a genetics expert who works in *Drosophila* embryogenesis. I do not have the expertise to evaluate the aggregate methods presented in Figure 4.

3. How much time do you estimate the authors will need to complete the suggested revisions:

Estimated time to Complete Revisions (Required)

(Decision Recommendation)

Less than 1 month

4. Review Commons values the work of reviewers and encourages them to get credit for their work. Select 'Yes' below to register your reviewing activity at Web of Science Reviewer Recognition Service (formerly Publons); note that the content of your review will not be visible on Web of Science.

Yes

Review #2

1. Evidence, reproducibility and clarity:

Evidence, reproducibility and clarity (Required)

****Summary:****

This manuscript focuses on the role of the histone chaperone NASP in *Drosophila*. NASP is a chaperone specific to histone H3 that is conserved in mammals. Many aspects of the molecular mechanisms by which NASP selectively binds histone H3 have been revealed through biochemical studies. However, key aspects of NASP's *in vivo* roles remain unclear, including where in the cell NASP functions, and how it prevents H3 degradation. Through live imaging in the early *Drosophila* embryo, which possesses large amounts of soluble H3 protein, Das et al determine that NASP does not control nuclear import or export of H3.2 or H3.3. Instead, they find through differential centrifugation analysis that NASP functions in the cytoplasm to prevent H3 aggregation and hence its subsequent degradation.

****Major Comments:****

1. The protein aggregation assays raise several questions.

a. From a technical standpoint, it would be helpful to have a positive control to demonstrate that the assay is effective at detecting protein aggregates. I.e. a genotype that exhibits increased protein aggregation; this could be for a protein besides H3.

b. If NASP is not required to prevent H3 degradation in egg chambers, then why are H3 levels much lower in NASP input lanes relative to wild-type egg chambers in Fig 4D?

c. A corollary to this is that the increased fraction of H3 in aggregates in NASP mutants seems to be entirely due to the reduction in total H3 levels rather than an increase in aggregated H3. If NASP's role is to prevent aggregation in the cytoplasm, and degradation has not yet begun in egg chambers, then why are aggregated H3 levels not increased in NASP mutants relative to wild-type egg chambers? If the same number of egg chambers were used, shouldn't the total amount of histone be the same in the absence of degradation?

2. The live imaging studies are well designed, executed, and quantified. They use an established genotype (H3.2-Dendra2) in wild-type and NASP maternal mutants to demonstrate that NASP is not directly involved in nuclear import of H3.2. Decreased import is likely due to reduced H3.2 levels in NASP mutants rather than reduced import rates per se. The same methodology was used to determine that loss of NASP did not affect H3.2 nuclear export. These findings eliminate H3.2 nuclear import/export regulation as possible roles for NASP, which had been previously proposed.
3. Live imaging also conclusively demonstrates that the levels of H3.2 in the nucleoplasm and in mitotic chromatin are significantly lower in NASP mutants than wild-type nuclei. Despite these lower histone levels, the nuclear cycle duration is only modestly lengthened.
4. The live imaging of NASP-Dendra2 nuclear import conclusively demonstrate that NASP and H3.2 are unlikely to be imported into the nucleus as one complex.

****Minor Comments:****

1. Additional details on how the NASP-Dendra2 CRISPR allele was generated should be provided. In addition, additional details on how it was determined that this allele is functional should be provided (e.g. quantitative assays for fertility/embryo viability of NASP-Dendra2 females)
2. If statistical tests are used to determine significance, the type of test used should be reported in the figure legends throughout.
3. The western blot shown in Figure 4A looks more like a 4-fold reduction in H3 levels in NASP mutants relative to wild-type embryos, rather than the quantified 2-fold reduction. Perhaps a more representative blot can be shown.

2. Significance:

Significance (Required)

As a fly chromatin biologist with colleagues that utilize mammalian experimental systems, I feel this manuscript will be of broad interest to the chromatin research community. Packaging of the genome into chromatin affects nearly every DNA-templated process, making the mechanisms by which histone proteins are expressed, chaperoned, and deposited into chromatin of high importance to the field. The study has multiple strengths, including high-quality quantitative imaging, use of a terrific experimental system (storage and deposition of soluble histones in early fly embryos). The study also answers outstanding questions in the field, specifically that NASP does not control nuclear import/export of histone H3. Instead, the authors propose that NASP functions to prevent protein aggregation. If this could be conclusively demonstrated, it would be valuable to the

field. However, the protein aggregation studies need improvement. Technical demonstration that their differential centrifugation assay accurately detects aggregated proteins is needed. Further, NASP mutants do not exhibit increased H3 protein aggregation in the data presented. Instead, the increased fraction of aggregated H3 in NASP mutants seems to be due to a reduction in the overall levels of H3 protein, which is contrary to the model presented in this paper.

3. How much time do you estimate the authors will need to complete the suggested revisions:

Estimated time to Complete Revisions (Required)

(Decision Recommendation)

Between 1 and 3 months

4. Review Commons values the work of reviewers and encourages them to get credit for their work. Select 'Yes' below to register your reviewing activity at Web of Science Reviewer Recognition Service (formerly Publons); note that the content of your review will not be visible on Web of Science.

No

Review #3

1. Evidence, reproducibility and clarity:

Evidence, reproducibility and clarity (Required)

This manuscript by Das et al. entitled "NASP functions in the cytoplasm to prevent histone H3 aggregation during early embryogenesis", explores the role of the histone chaperone NASP in regulating histone H3 dynamics during early Drosophila embryogenesis. Using primarily live imaging approaches, the authors found that NASP is not directly involved in the import or export of H3. Moreover, the authors claimed that NASP prevents H3 aggregation rather than protects against degradation.

****Major Comments:****

Figure 1A-B: The plotted data appear to have substantial dispersion. Could the authors include individual data points or provide representative images to help the reader assess

variability?

Given that the authors conclude that the reduced nuclear import is due to lowered H3 levels in NASP-deficient embryos, would overexpression of H3 rescue this phenotype? This would directly test whether H3 levels, rather than import machinery per se, drive the effect.

Figure 2A-B: The authors present the Relative Intensity of H3-Dendra2, but this metric obscures absolute differences between Control and NASP knockout embryos. Please include Total Intensity plots to show the actual reduction in H3 levels.

Additionally, Western blot analysis of nucleoplasmic H3 from wild-type vs. NASP-deficient embryos would provide essential biochemical confirmation of H3 level reductions.

Figure 4: To support the conclusion that NASP prevents H3 aggregation, I recommend performing aggregation assays by adding compounds that induce unfolding (amino acid analogues that induce unfolding, like canavanine or Azetidine-2-carboxylic acid) or using aggregation-prone H3 mutants.

Inclusion of CMA and proteasome inhibition experiments could also clarify whether degradation pathways are secondarily involved or compensatory in the absence of NASP.

****Minor Comments:****

(1) The Introduction would benefit from mentioning the two NASP isoforms that exist in mammals (sNASP and tNASP), as this evolutionary context may inform interpretation of the *Drosophila* results.

(2) Could the authors comment on the status of histone H4 in their experimental system? Given the observed cytoplasmic pool of H3, is it likely to exist as a monomer? If this H3 pool is monomeric, does that suggest an early failure in H3-H4 dimerization, and could this contribute to its aggregation propensity?

2. Significance:

Significance (Required)

This work addresses a timely and important question in the field of chromatin biology and developmental epigenetics. The focus on histone homeostasis during embryogenesis and the cytoplasmic role of NASP adds a novel perspective. The live imaging experiments are a clear strength, providing valuable spatiotemporal insights. However, I believe that the manuscript would benefit significantly from additional biochemical validation to support and clarify some of the mechanistic claims.

3. How much time do you estimate the authors will need to complete the suggested revisions:

Estimated time to Complete Revisions (Required)

(Decision Recommendation)

Between 1 and 3 months

Yes

Revision Plan

Manuscript number: RC-2025-03208

Corresponding author(s): Jared Nordman

[The “revision plan” should delineate the revisions that authors intend to carry out in response to the points raised by the referees. It also provides the authors with the opportunity to explain their view of the paper and of the referee reports.]

The document is important for the editors of affiliate journals when they make a first decision on the transferred manuscript. It will also be useful to readers of the reprint and help them to obtain a balanced view of the paper.

*If you wish to submit a full revision, please use our "Full Revision" template. **It is important to use the appropriate template to clearly inform the editors of your intentions.**]*

1. General Statements [optional]

All three reviewers of our manuscript were very positive about our work. The reviewers noted that our work represents a necessary advance that is timely, addresses important issues in the chromatin field, and will of broad interest to this community. Given the nature of our work and the positive reviews, we feel that this manuscript would best be suited for the *Journal of Cell Biology*.

2. Description of the planned revisions

Reviewer #1 (Evidence, reproducibility and clarity (Required)):

Summary:

The authors investigate the function of the H3 chaperone NASP, which is known to bind directly to H3 and prevent degradation of soluble H3. What is unclear is where NASP functions in the cell (nucleus or cytoplasm), how NASP protects H3 from degradation (direct or indirect), and if NASP affects H3 dynamics (nuclear import or export). They use the powerful model system of *Drosophila* embryos because the soluble H3 pool is high due to maternal deposition and they make use of photoconvertible Dendra-tagged proteins, since these are maternally deposited and can be used to measure nuclear import/export rates.

Using these systems and tools, they conclude that NASP affects nuclear import, but only indirectly, because embryos from NASP mutant mothers start out with 50% of the maternally deposited H3. Because of the depleted H3 and reduced import rates, NASP deficient embryos also have reduced nucleoplasmic and chromatin-associated H3. Using a new Dendra-tagged NASP allele, the authors show that NASP and H3 have different nuclear import rates, indicating

Revision Plan

that NASP is not a chaperone that shuttles H3 into the nucleus. They test H3 levels in embryos that have no nuclei and conclude that NASP functions in the cytoplasm, and through protein aggregation assays they conclude that NASP prevents H3 aggregation.

Major comments:

The text was easy to read and logical. The data are well presented, methods are complete, and statistics are robust. The conclusions are largely reasonable. However, I am having trouble connecting the conclusions in text to the data presented in Figure 4.

First, I'm confused why the conclusion from Figure 4A is that NASP functions in the cytoplasm of the egg. Couldn't NASP be required in the ovary (in, say, nurse cell nuclei) to stimulate H3 expression and deposition into the egg? The results in 4A would look the same if the mothers deposit 50% of the normal H3 into the egg. Why is NASP functioning specifically in the cytoplasm when it is also so clearly imported into the nucleus? Maybe NASP functions wherever it is, and by preventing nuclear import, you force it to function in the cytoplasm. I do not have additional suggestions for experiments, but I think the authors need to be very clear about the different interpretations of these data and to discuss WHY they believe their conclusion is strongest.

The concern raised by the reviewer regarding NASP function during oogenesis has been addressed in a previous work published from our lab. Unfortunately, we did not do a good job conveying this work in the original version of this manuscript. We demonstrated that total H3 levels are unaffected when comparing WT and *NASP* mutant stage 14 egg chambers. This means that the amount of H3 deposited into the eggs does not change in the absence of *NASP*. To address the reviewer's comment, we will change the text to make the link to our previous work clear.

Second, an alternate conclusion from Figure 4D/E is that mothers are depositing less H3 protein into the egg, but the same total amount is being aggregated. This amount of aggregated protein remains constant in activated eggs, but additional H3 translation leads to more total H3? The authors mention that additional translation can compensate for reduced histone pools (line 416).

Similar to our response above, the total amount of H3 in wild type and *NASP* mutant stage 14 egg chambers is the same. Therefore, mothers are depositing equal amounts of H3 into the egg. We will make the necessary changes in the text to make this point clear.

As the function of *NASP* in the cytoplasm (when it clearly imports into the nucleus) and role in H3 aggregation are major conclusions of the work, the authors need to present alternative conclusions in the text or complete additional experiments to support the claims. Again, I do not have additional suggestions for experiments, but I think the authors need to be very clear about the different interpretations of these data and to discuss WHY they believe their conclusion is strongest.

Revision Plan

A common issue raised by all three reviewers was to more convincingly demonstrate that assay that we have used to isolate protein aggregates does, in fact, isolate protein aggregates. To verify this, we will be performing the aggregate isolation assay using controls that are known to induce more protein aggregation. We will perform the aggregation assay with egg chambers or extracts that are exposed to heat shock or the aggregation-inducing chemicals Canavanine and Azetidine-2-carboxylic acid. The chemical treatment was a welcome suggestion from reviewer #3. These experiments will significantly strengthen any claims based on the outcome of the aggregation assay.

We will also make changes to the text and include other interpretations of our work as the reviewer has suggested.

Data presentation:

Overall, I suggest moving some of the supplemental figures to the main text, adding representative movie stills to show where the quantitative data originated, and moving the H3.3 data to the supplement. Not because it's not interesting, but because H3.3 and H3.2 are behaving the same.

Where possible, we will make changes to the figure display to improve the logic and flow of the manuscript

Fig 1:

It would strengthen the figure to include representative still images that led to the quantitative data, mostly so readers understand how the data were collected.

We will add representative stills to Figure 1 to help readers understand how the data is collected. We will also add a representative H3-Dendra movie similar to the NASP supplemental movie.

The inclusion of a "simulated 50% H3" in panel C is confusing. Why?

We used a 50% reduction in H3 levels because that is reduction in H3 we measure in embryos laid by NASP-mutant mothers in our previous work. A reduction in H3 levels alone would be predicted to change the nuclear import rate of H3. Thus, having a quantitative model of H3 import kinetics was key in our understanding of NASP function in vivo. We will revise the text to make this clear.

I would also consider normalizing the data between A and B (and C and D) by dividing NASP/WT. This could be included in the supplement (OPTIONAL)

We can normalize the values and include the data in a supplemental figure.

Revision Plan

Fig S1:

The data simulation S1G should be moved to the main text, since it is the primary reason the authors reject the hypothesis that NASP influences H3 import rates.

This is a good point. We will move S1G into the Figure 1.

Fig 2:

Once again, I think it would help to include a few representative images of the photoconverted Dendra2 in the main text.

We will add representative images of the photoconversion in Figure 2.

I struggled with A/B, I think due to not knowing how the data were normalized. When I realized that the WT and NASP data are not normalized to each other, but that the NASP values are likely starting less than the WT values, it made way more sense. I suggest switching the order of data presentation so that C-F are presented first to establish that there is less chromatin-bound H3 in the first place, and then present A/B to show no change in nuclear export of the H3 that is present, allowing the conclusion of both less soluble AND chromatin-bound H3.

The order of the presentation of the data was to test if NASP was acting as a nuclear receptor. Since Figure 1 compares the nuclear import, we wanted to address the nuclear export and provide a comprehensive analysis of the role of NASP in H3 nuclear dynamics before advancing on to other consequences of NASP depletion. We can add the graphs with the un-normalized values in the Supplemental Figure to show the actual difference in total intensity values.

Fig S2:

If M1-M3 indicate males, why are the ovaries also derived from males? I think this is just confusing labeling.

We will change the labelling.

Supplemental Movie S1:

Beautiful. Would help to add a time stamp (OPTIONAL).

Thank you! We will add the time stamp to the movie

Fig 3:

Panel C is the same as Fig S1A (not Fig 1A, as is said in the legend), though I appreciate the authors pointing it out in the legend. Also see line 276.

Revision Plan

We appreciate the reviewer for pointing this out. We will make the change in the text to correct this.

Panel D is a little confusing, because presumably the "% decrease in import rate" cannot be positive (Y axis). This could be displayed as a scatter (not bar) as in Panels B/C (right) where the top of the Y axis is set to 0.

We understand the reviewer's concern that the decrease value cannot be positive. We can adjust the y-axis so that it caps off at 0.

Fig S3:

A: What do the different panels represent? I originally thought developmental time, but now I think just different representative images? Are these age-matched from time at egg lay?

The different panels show representative images. We can clarify that in the figure legend.

C: What does "embryos" mean? Same question for Fig 4A.

In this figure, embryos mean the exact number of embryos used to form the lysate for the western blot. We will clarify this in the figure legend.

Fig 4:

A: What does "embryos" mean? Number of embryos? Age in hours?

In this figure, embryos mean the exact number of embryos used to form the lysate for the western blot. We will clarify this in the figure legend.

C: Not sure the workflow figure panel is necessary, as I can't tell what each step does. This is better explained in methods. However I appreciated the short explanation in the text (lines 314-5).

The workflow panel helps to identify the samples labelled as input and aggregate for the western blot analysis. Since our input in the western blots does not refer to the total protein lysate, we feel it is helpful to point out exactly what stage at the protocol we are utilizing the sample for our analysis.

Minor comments:

The authors should describe the nature of the NASP alleles in the main text and present evidence of robust NASP depletion, potentially both in ovaries and in embryos. The antibody works well for westerns (Fig S2B). This is sort of demonstrated later in Figure 4A, but only in NAAP x twine activated eggs.

Revision Plan

We appreciate the reviewer's comments about the NASP mutant allele. In our previous publication, we characterized the NASP mutant fly line and its effect on both stage 14 egg chambers and the embryos. We will emphasize the reference to our previous work in the text.

Lines 163, 251, 339: minor typos

Line 184: It would help to clarify- I'm assuming cytoplasmic concentration (or overall) rather than nuclear concentration. If nuclear, I'd expect the opposite relationship. This occurs again when discussing NASP (line 267). I suspect it's also not absolute concentration, but relative concentration difference between cytoplasm and nucleus. It would help clarify if the authors were more precise.

We appreciate the reviewer's point and will add the clarification in the text.

Line 189: Given that the "established integrative model" helps to reject the hypothesis that NASP is involved in H3 import, I think it's important to describe the model a little more, even though it's previously published.

We will add few sentences giving a brief description of the model to the text.

Line 203: "The measured rate of H3.2 export from the nucleus is negligible" clarify this is in WT situations and not a conclusion from this study.

We will add the clarification of this statement in the text.

Line 211: How can the authors be so sure that the decrease in WT is due to "the loss of non-chromatin bound nucleoplasmic H3.2-Dendra2?"

From the live imaging experiments, the H3.2-Dendra2 intensity in the nucleus reduces dramatically upon nuclear envelope breakdown with the only H3.2-Dendra2 intensity remaining being the chromatin bound H3.2. Excess H3.2 is imported into the nucleus and not all of it is incorporated into the chromatin. This is a unique feature of the embryo system that has been observed previously. We mention that the intensity reduction is due to the loss of non-chromatin bound nucleoplasmic H3.2.

Line 217: In the conclusion, the authors indicate that NASP indirectly affects soluble supply of H3 in the nucleoplasm. I do believe they've shown that the import rate effect is indirect, but I don't know why they conclude that the effect of NASP on the soluble nucleoplasmic H3 supply is indirect. Similarly, the conclusion is indirect on line 239. Yet, the authors have not shown it's not direct, just assumed since NASP results in 50% decrease to deposited maternal histones.

We appreciate the feedback on the conclusions of Figure 2 from the reviewer. Our conclusions are primarily based on the effect of H3 levels in the absence of NASP in the early embryos. To establish direct causal effects, it would be important to recover the phenotypes by

Revision Plan

complementation experiments and providing molecular interactions to cause the effects. In this study we have not established those specific details to make conclusions of direct effects. We will change the text to make this more clear.

Line 292: What is the nature of the NASP "mutant?" Is it a null? Similarly, what kind of "mutant" is the twine allele? Line 295.

We will include descriptions of the NASP and twine mutants in the text.

Line 316: Why did the authors use stage 14 egg chambers here when they previously used embryos? This becomes more clear later shortly, when the authors examine activated eggs, but it's confusing in text.

The reason to use stage 14 egg chambers was to establish NASP function during oogenesis. We will modify the text to emphasize the reason behind using stage 14 egg chambers.

Lines 343-348: It's unclear if the authors are drawing extended conclusions here or if they are drawing from prior literature (if so, citations would be required). For example, why during oogenesis/embryogenesis are aggregation and degradation developmentally separated?

This conclusion is based primarily based on the findings from this study (Figure 4) and out previous published work. We will modify the text for more clarity.

Lines 386-7: I do not understand why the authors conclude that H3 aggregation and degradation are "developmentally uncoupled" and why, in the absence of NASP, "H3 aggregation precedes degradation."

This is based data in Figure 4 combined with our previous working showing that the total level of H3 in not changed in NASP-mutant stage 14 egg chambers. Aggregates seem to be more persistent in the stage 14 egg chambers (oogenesis) and they get cleared out upon egg activation (entry into embryogenesis). This provides evidence for aggregation occurring prior to degradation and these two events occurring in different developmental stages. We will change the text to make this more clear.

Line 395: Why suddenly propose that NASP also functions in the nucleus to prevent aggregation, when earlier the authors suggest it functions only in the cytoplasm?

We will make the necessary edits to ensure that the results don't suggest a role of NASP exclusive to the cytoplasm. Our findings highlight a cytoplasmic function of NASP, however, we do not want to rule out that this same function couldn't occur in the nucleus.

Lines 409-413: The authors claim that histone deficiency likely does not cause the embryonic arrest seen in embryos from NASP mutant mothers. This is because H3 is reduced by 50% yet

Revision Plan

some embryos arrest long before they've depleted this supply. However, the authors also showed that H3 import rates are affected in these embryos due to lower H3 concentration. Since the early embryo cycles are so rapid, reduced H3 import rates could lead to early arrest, even though available H3 remains in the cytoplasm.

We thank the reviewer for their suggestion. This conclusion is based on the findings from the previous study from our lab which showed that the majority of the embryos laid by NASP mutant females get arrested in the very early nuclear cycles (<NC5) whereas our live imaging studies show that NASP mutant embryos are capable of progressing into later nuclear cycles even with significantly reduced H3 levels. Thus, we conclude that H3 reduction is not sufficient to explain the defects observed in NASP mutant embryos.

Reviewer #1 (Significance (Required)):

The significance of the work is conceptual, as NASP is known to function in H3 availability but the precise mechanism is elusive. This work represents a necessary advance, especially to show that NASP does not affect H3 import rates, nor does it chaperone H3 into the nucleus. However, the authors acknowledge that many questions remain. Foremost, why is NASP imported into the nucleus and what is its role there?

I believe this work will be of interest to those who focus on early animal development, but NASP may also represent a tool, as the authors conclude in their discussion, to reduce histone levels during development and examine nucleosome positioning. This may be of interest to those who work on chromatin accessibility and zygotic genome activation.

I am a genetics expert who works in Drosophila embryogenesis. I do not have the expertise to evaluate the aggregate methods presented in Figure 4.

Reviewer #2 (Evidence, reproducibility and clarity (Required)):

Summary:

This manuscript focuses on the role of the histone chaperone NASP in Drosophila. NASP is a chaperone specific to histone H3 that is conserved in mammals. Many aspects of the molecular mechanisms by which NASP selectively binds histone H3 have been revealed through biochemical studies. However, key aspects of NASP's in vivo roles remain unclear, including where in the cell NASP functions, and how it prevents H3 degradation. Through live imaging in the early Drosophila embryo, which possesses large amounts of soluble H3 protein, Das et al determine that NASP does not control nuclear import or export of H3.2 or H3.3. Instead, they find through differential centrifugation analysis that NASP functions in the cytoplasm to prevent H3 aggregation and hence its subsequent degradation.

Revision Plan

Major Comments:

1. The protein aggregation assays raise several questions.

a. From a technical standpoint, it would be helpful to have a positive control to demonstrate that the assay is effective at detecting protein aggregates. I.e. a genotype that exhibits increased protein aggregation; this could be for a protein besides H3.

A common issue raised by all three reviewers was to more convincingly demonstrate that assay that we have used to isolate protein aggregates does, in fact, isolate protein aggregates. To verify this, we will be performing the aggregate isolation assay using controls that are known to induce more protein aggregation. We will perform the aggregation assay with egg chambers or extracts that are exposed to heat shock or the aggregation-inducing chemicals Canavanine and Azetidine-2-carboxylic acid. The chemical treatment was a welcome suggestion from reviewer #3. These experiments will significantly strengthen any claims based on the outcome of the aggregation assay.

b. If NASP is not required to prevent H3 degradation in egg chambers, then why are H3 levels much lower in NASP input lanes relative to wild-type egg chambers in Fig 4D?

We appreciate the reviewer's inputs regarding the reduced H3 levels in the NASP mutant egg chambers. We observe this reduction in H3 levels in the input because of the altered solubility of H3 which leads to the loss of H3 protein at different steps of the aggregate isolation assay. We will add a supplement figure showing H3 levels at different steps of the aggregate isolation assay. We do want to stress, however, that the total levels of H3 in stage 14 egg chambers does not change between WT and the *NASP* mutant.

c. A corollary to this is that the increased fraction of H3 in aggregates in NASP mutants seems to be entirely due to the reduction in total H3 levels rather than an increase in aggregated H3. If NASP's role is to prevent aggregation in the cytoplasm, and degradation has not yet begun in egg chambers, then why are aggregated H3 levels not increased in NASP mutants relative to wild-type egg chambers? If the same number of egg chambers were used, shouldn't the total amount of histone be the same in the absence of degradation?

In previously published work, we demonstrated that total H3 levels are unaffected when comparing WT and *NASP* mutant stage 14 egg chambers. This means that the amount of H3 deposited into the eggs does not change in the absence of NASP. To address the reviewer's comment, we will change the text to make the link to our previous work clear. As stated above, we will add a supplement figure showing H3 levels at different steps of the aggregate isolation assay.

2. The live imaging studies are well designed, executed, and quantified. They use an established genotype (H3.2-Dendra2) in wild-type and NASP maternal mutants to demonstrate that NASP is not directly involved in nuclear import of H3.2. Decreased import is likely due to

Revision Plan

reduced H3.2 levels in NASP mutants rather than reduced import rates per se. The same methodology was used to determine that loss of NASP did not affect H3.2 nuclear export. These findings eliminate H3.2 nuclear import/export regulation as possible roles for NASP, which had been previously proposed.

Thank you.

3. Live imaging also conclusively demonstrates that the levels of H3.2 in the nucleoplasm and in mitotic chromatin are significantly lower in NASP mutants than wild-type nuclei. Despite these lower histone levels, the nuclear cycle duration is only modestly lengthened.
4. The live imaging of NASP-Dendra2 nuclear import conclusively demonstrate that NASP and H3.2 are unlikely to be imported into the nucleus as one complex.

Thank you.

Minor Comments:

1. Additional details on how the NASP-Dendra2 CRISPR allele was generated should be provided. In addition, additional details on how it was determined that this allele is functional should be provided (e.g. quantitative assays for fertility/embryo viability of NASP-Dendra2 females)

We will make these additions to the text.

2. If statistical tests are used to determine significance, the type of test used should be reported in the figure legends throughout.

We will make the addition of the statistical tests to the figure legends.

3. The western blot shown in Figure 4A looks more like a 4-fold reduction in H3 levels in NASP mutants relative to wild-type embryos, rather than the quantified 2-fold reduction. Perhaps a more representative blot can be shown.

We have additional blots in the supplemental figure S3C. The quantification was performed after normalization to the total protein levels and we can highlight that in the figure legend.

Reviewer #2 (Significance (Required)):

As a fly chromatin biologist with colleagues that utilize mammalian experimental systems, I feel this manuscript will be of broad interest to the chromatin research community. Packaging of the genome into chromatin affects nearly every DNA-templated process, making the mechanisms by which histone proteins are expressed, chaperoned, and deposited into chromatin of high importance to the field. The study has multiple strengths, including high-quality quantitative

Revision Plan

imaging, use of a terrific experimental system (storage and deposition of soluble histones in early fly embryos). The study also answers outstanding questions in the field, specifically that NASP does not control nuclear import/export of histone H3. Instead, the authors propose that NASP functions to prevent protein aggregation. If this could be conclusively demonstrated, it would be valuable to the field. However, the protein aggregation studies need improvement. Technical demonstration that their differential centrifugation assay accurately detects aggregated proteins is needed. Further, NASP mutants do not exhibit increased H3 protein aggregation in the data presented. Instead, the increased fraction of aggregated H3 in NASP mutants seems to be due to a reduction in the overall levels of H3 protein, which is contrary to the model presented in this paper.

Reviewer #3 (Evidence, reproducibility and clarity (Required)):

This manuscript by Das et al. entitled "NASP functions in the cytoplasm to prevent histone H3 aggregation during early embryogenesis", explores the role of the histone chaperone NASP in regulating histone H3 dynamics during early *Drosophila* embryogenesis. Using primarily live imaging approaches, the authors found that NASP is not directly involved in the import or export of H3. Moreover, the authors claimed that NASP prevents H3 aggregation rather than protects against degradation.

Major Comments:

Figure 1A-B: The plotted data appear to have substantial dispersion. Could the authors include individual data points or provide representative images to help the reader assess variability?

We chose to show unnormalized data in Figure 1 so readers could better compare the actual import values of H3 in the presence and absence of NASP. We felt it was a better representation of the true biological difference although raw data is more dispersive. We did also include normalized data in the supplement. Regardless, we will add representative stills to Figure 1 and include a H3-Dendra2 movie in the supplement to show the representative data.

Given that the authors conclude that the reduced nuclear import is due to lowered H3 levels in NASP-deficient embryos, would overexpression of H3 rescue this phenotype? This would directly test whether H3 levels, rather than import machinery per se, drive the effect.

We thank the reviewer for their valuable suggestion. We and others have tried to overexpress histones in the *Drosophila* early embryo without success. There must be an undefined feedback mechanism preventing histone overexpression in the germline. In fact, a recent paper has been deposited on *bioRxiv* (<https://doi.org/10.1101/2024.12.23.630206>) that suggest H4 protein could provide a feedback mechanism to prevent histone overexpression. While we would love to do this experiment, it is not technically feasible at this time.

Revision Plan

Figure 2A-B: The authors present the Relative Intensity of H3-Dendra2, but this metric obscures absolute differences between Control and NASP knockout embryos. Please include Total Intensity plots to show the actual reduction in H3 levels.

We will add the total H3-Dendra2 intensity plots to the supplemental figure for the export curves.

Additionally, Western blot analysis of nucleoplasmic H3 from wild-type vs. NASP-deficient embryos would provide essential biochemical confirmation of H3 level reductions.

We will measure nuclear H3 levels by western from 0-2 hr embryos laid by WT and NASP mutant flies.

Figure 4: To support the conclusion that NASP prevents H3 aggregation, I recommend performing aggregation assays by adding compounds that induce unfolding (amino acid analogues that induce unfolding, like canavanine or Azetidine-2-carboxylic acid) or using aggregation-prone H3 mutants.

This is a very helpful suggestion! It is difficult to get chemicals into *Drosophila* eggs, but we will treat extracts directly with these chemicals. Additionally, we will use heat shocked eggs and extracts as an additional control.

Inclusion of CMA and proteasome inhibition experiments could also clarify whether degradation pathways are secondarily involved or compensatory in the absence of NASP.

The degradation pathway for H3 in the absence of NASP is unknown and a major focus of our future work is to define this pathway. *Drosophila* does not have a CMA pathway and therefore, we don't know how H3 aggregates are being sensed.

Minor Comments:

(1) The Introduction would benefit from mentioning the two NASP isoforms that exist in mammals (sNASP and tNASP), as this evolutionary context may inform interpretation of the *Drosophila* results.

We will make the edits in the text to include that *Drosophila* NASP is the sole homolog of sNASP and that tNASP ortholog is not found in *Drosophila*.

(2) Could the authors comment on the status of histone H4 in their experimental system? Given the observed cytoplasmic pool of H3, is it likely to exist as a monomer? If this H3 pool is monomeric, does that suggest an early failure in H3-H4 dimerization, and could this contribute to its aggregation propensity?

In our previous work we noted that NASP binds more preferentially to H3 and the levels of H3 we much more reduced upon NASP depletion than H4. We pointed out in this publication that

Revision Plan

our data was consistent with H3 stores being monomeric in the *Drosophila* embryo. We don't have a H4-Dendra2 line to test. In the future, however, this is something we are very keen to look at.

Reviewer #3 (Significance (Required)):

This work addresses a timely and important question in the field of chromatin biology and developmental epigenetics. The focus on histone homeostasis during embryogenesis and the cytoplasmic role of NASP adds a novel perspective. The live imaging experiments are a clear strength, providing valuable spatiotemporal insights. However, I believe that the manuscript would benefit significantly from additional biochemical validation to support and clarify some of the mechanistic claims.

3. Description of the revisions that have already been incorporated in the transferred manuscript

4. Description of analyses that authors prefer not to carry out

Please include a point-by-point response explaining why some of the requested data or additional analyses might not be necessary or cannot be provided within the scope of a revision. This can be due to time or resource limitations or in case of disagreement about the necessity of such additional data given the scope of the study. Please leave empty if not applicable.

December 16, 2025

Re: JCB manuscript #202511182T

Jared Nordman
Vanderbilt University

Dear Dr. Nordman,

Thank you for submitting your Transfer manuscript entitled "NASP functions in the cytoplasm to prevent histone H3 aggregation during early embryogenesis" via Review Commons. Based on the reviewer feedback and our editorial assessment, we invite you to submit a revised manuscript based on the revision plan you provided.

We agree with the reviewers, thinking that your manuscript is interesting and addresses a long-standing question concerning histone metabolism and the function of chaperone proteins. The reviewers provide constructive feedback, and we think your paper could improve if you were able to address all of their concerns. This includes additional experiments to strengthen the conclusions derived from the aggregation assay.

GENERAL GUIDELINES:

Text limits: Character count for an Transfer is < 40,000, not including spaces. Count includes title page, abstract, introduction, results, discussion, and acknowledgments. Count does not include materials and methods, figure legends, references, tables, or supplemental legends.

Figures: Transfers may have up to 10 main text figures. Figures must be prepared according to the policies outlined in our Instructions to Authors, under Data Presentation, <https://jcb.rupress.org/site/misc/ifora.xhtml>. All figures in accepted manuscripts will be screened prior to publication.

*****IMPORTANT:** It is JCB policy that if requested, original data images must be made available. Failure to provide original images upon request will result in unavoidable delays in publication. Please ensure that you have access to all original microscopy and blot data images before submitting your revision. ***

Supplemental information: There are strict limits on the allowable amount of supplemental data. Transfers may have up to 5 supplemental figures. Up to 10 supplemental videos or flash animations are allowed. A summary of all supplemental material should appear at the end of the Materials and methods section.

Please note that JCB now requires authors to submit Source Data used to generate figures containing gels and Western blots with all revised manuscripts. This Source Data consists of fully uncropped and unprocessed images for each gel/blot displayed in the main and supplemental figures. For assays performed using capillary electrophoresis and/or immunoassay-based detection, authors should instead provide the electropherogram graph(s) for each experiment, plotting fluorescence/chemiluminescence intensity vs. molecular weight/size. Please be sure to provide one Source Data file for each figure gels, blots, and/or capillary electrophoresis assays along with your revised manuscript files. File names for Source Data figures should be alphanumeric without any spaces or special characters (i.e., SourceDataF#, where F# refers to the associated main figure number or SourceDataFS# for those associated with Supplementary figures). For traditional gels and blots, the lanes of the gels/blots should be labeled as they are in the associated figure, the place where cropping was applied should be marked (with a box), and molecular weight/size standards should be labeled wherever possible. For capillary electrophoresis assays, each trace in the graph should be color-coded and labeled to indicate which protein, gene, or sample is being measured (please try to avoid red/green combinations to accommodate our color-blind readers).

The typical timeframe for revisions is three to four months. If you anticipate any difficulties in meeting this aforementioned revision time limit, please contact us and we can work with you to find an appropriate time frame for resubmission. Please note that papers are generally considered through only one revision cycle, so any revised manuscript will likely be either accepted or

rejected.

Thank you for this interesting contribution to Journal of Cell Biology. You can contact us at the journal office with any questions at cellbio@rockefeller.edu.

Sincerely,

Hiroshi Kimura
Monitoring Editor
Journal of Cell Biology

Gabriele Stephan
Scientific Editor
Journal of Cell Biology

Reviewer #1 (Evidence, reproducibility and clarity (Required)):

Summary:

The authors investigate the function of the H3 chaperone NASP, which is known to bind directly to H3 and prevent degradation of soluble H3. What is unclear is where NASP functions in the cell (nucleus or cytoplasm), how NASP protects H3 from degradation (direct or indirect), and if NASP affects H3 dynamics (nuclear import or export). They use the powerful model system of *Drosophila* embryos because the soluble H3 pool is high due to maternal deposition and they make use of photoconvertible Dendra-tagged proteins, since these are maternally deposited and can be used to measure nuclear import/export rates.

Using these systems and tools, they conclude that NASP affects nuclear import, but only indirectly, because embryos from NASP mutant mothers start out with 50% of the maternally deposited H3. Because of the depleted H3 and reduced import rates, NASP deficient embryos also have reduced nucleoplasmic and chromatin-associated H3. Using a new Dendra-tagged NASP allele, the authors show that NASP and H3 have different nuclear import rates, indicating that NASP is not a chaperone that shuttles H3 into the nucleus. They test H3 levels in embryos that have no nuclei and conclude that NASP functions in the cytoplasm, and through protein aggregation assays they conclude that NASP prevents H3 aggregation.

Major comments:

The text was easy to read and logical. The data are well presented, methods are complete, and statistics are robust. The conclusions are largely reasonable. However, I am having trouble connecting the conclusions in text to the data presented in Figure 4.

First, I'm confused why the conclusion from Figure 4A is that NASP functions in the cytoplasm of the egg. Couldn't NASP be required in the ovary (in, say, nurse cell nuclei) to stimulate H3 expression and deposition into the egg? The results in 4A would look the same if the mothers deposit 50% of the normal H3 into the egg. Why is NASP functioning specifically in the cytoplasm when it is also so clearly imported into the nucleus? Maybe NASP functions wherever it is, and by preventing nuclear import, you force it to function in the cytoplasm. I do not have additional suggestions for experiments, but I think the authors need to be very clear about the different interpretations of these data and to discuss WHY they believe their conclusion is strongest.

The concern raised by the reviewer regarding NASP function during oogenesis has been addressed in a previous work published from our lab. Unfortunately, we did not do a good job conveying this work in the original version of this manuscript. We demonstrated that total H3 levels are unaffected when comparing WT and *NASP* mutant stage 14 egg chambers. This means that the amount of H3 deposited into the eggs does not change in the absence of NASP. We have referenced our previous work more clearly and added a western blot in Figure 4 showing that H3 levels in WT and *NASP* stage 14 egg chambers are unchanged while H3 levels are reduced only upon egg activation.

Second, an alternate conclusion from Figure 4D/E is that mothers are depositing less H3 protein into the egg, but the same total amount is being aggregated. This amount of aggregated protein remains constant in activated eggs, but additional H3 translation leads to more total H3? The authors mention that additional translation can compensate for reduced histone pools (line 416).

Similar to our response above, we previously published the total amount of H3 in wild type and *NASP* mutant stage 14 egg chambers is the same. Therefore, mothers are depositing equal amounts of H3 into the egg. We have referenced our previous work more clearly and added a western blot in Figure 4 showing that H3 levels in WT and *NASP* stage 14 egg chambers are unchanged while H3 levels are reduced only upon egg activation.

As the function of *NASP* in the cytoplasm (when it clearly imports into the nucleus) and role in H3 aggregation are major conclusions of the work, the authors need to present alternative conclusions in the text or complete additional experiments to support the claims. Again, I do not have additional suggestions for experiments, but I think the authors need to be very clear about the different interpretations of these data and to discuss WHY they believe their conclusion is strongest.

A common issue raised by all three reviewers was to more convincingly demonstrate that assay that we have used to isolate protein aggregates does, in fact, isolate protein aggregates. To verify this, we have performed the aggregate isolation assay using controls that are known to induce more protein aggregation.

We have performed the aggregate isolation assay on WT egg chambers with chemical that are known to induce aggregate formation as positive controls for aggregate formation (Canavanine and Azetidine-2-carboxylic acid). Additionally, we performed the aggregate isolation assay on S2 cells with Canavanine treatment to further validate the assay. Treatment of egg chambers and S2 cells with these chemicals increased aggregate formation as expected. We have included these critical controls in a revised Figure 5 and Figure S4. We have also modified the text accordingly.

Data presentation:

Overall, I suggest moving some of the supplemental figures to the main text, adding representative movie stills to show where the quantitative data originated, and moving the H3.3 data to the supplement. Not because it's not interesting, but because H3.3 and H3.2 are behaving the same.

We have made changes to multiple figures to improve the logic and flow of the manuscript.

Fig 1:

It would strengthen the figure to include representative still images that led to the quantitative data, mostly so readers understand how the data were collected.

We have included supplemental movies of the H3.2-Dendra2 and NASP-Dendra2 to provide a visual reference of nuclear import of the two proteins. Since, each frame is a multi-z-stack and images are taken over 240s time frame, it's hard to provide a still representation of the import analysis. We have now included the raw data for each of our embryos as excel spreadsheets. Raw image files will available upon reasonable request.

The inclusion of a "simulated 50% H3" in panel C is confusing. Why?

We used a 50% reduction in H3 levels because that is reduction in H3 we measure in embryos laid by NASP-mutant mothers in our previous work. We have revised the text to make this clear.

I would also consider normalizing the data between A and B (and C and D) by dividing NASP/WT. This could be included in the supplement (OPTIONAL)

The primary comparison for our conclusions come from the total intensities (used in Figure 1) and we have the normalized values for the same experiment in Supplemental Figure 1. Additional normalization of the NASP values to the WT values will not necessarily provide us any additional information. We chose to leave out this additional normalization to limit the number of curves and not confuse readers.

Fig S1:

The data simulation S1G should be moved to the main text, since it is the primary reason the authors reject the hypothesis that NASP influences H3 import rates.

The key simulated data is currently present in Figure 1. We tried to move the simulation H3 curves from the supplement, but it made the figure too busy and somewhat confusing to follow. Therefore, we decided to keep the simulation curves in the Supplemental Material.

Fig 2:

Once again, I think it would help to include a few representative images of the photoconverted Dendra2 in the main text.

We have added representative images showing the photoconversion of H3-Dendra2 with a time stamp in Figure 2A.

I struggled with A/B, I think due to not knowing how the data were normalized. When I realized that the WT and NASP data are not normalized to each other, but that the NASP values are likely starting less than the WT values, it made way more sense. I suggest switching the order of data presentation so that C-F are presented first to establish that there is less chromatin-bound

H3 in the first place, and then present A/B to show no change in nuclear export of the H3 that is present, allowing the conclusion of both less soluble AND chromatin-bound H3.

The order of the presentation of the data was to test if NASP was acting as a nuclear receptor. Since Figure 1 compares the nuclear import, we wanted to address the nuclear export and provide a comprehensive analysis of the role of NASP in H3 nuclear dynamics before advancing on to other consequences of NASP depletion. The order of this presentation follows the text. Thus, we decided to leave it as in the original manuscript.

Fig S2:

If M1-M3 indicate males, why are the ovaries also derived from males? I think this is just confusing labeling.

We have revised the supplemental figure and the legend.

Supplemental Movie S1:

Beautiful. Would help to add a time stamp (OPTIONAL).

We have added the time stamp to the movie

Fig 3:

Panel C is the same as Fig S1A (not Fig 1A, as is said in the legend), though I appreciate the authors pointing it out in the legend. Also see line 276.

We have made the change to the text to fix this.

Panel D is a little confusing, because presumably the "% decrease in import rate" cannot be positive (Y axis). This could be displayed as a scatter (not bar) as in Panels B/C (right) where the top of the Y axis is set to 0.

Thank you for pointing this out. We have fixed this in the figure.

Fig S3:

A: What do the different panels represent? I originally thought developmental time, but now I think just different representative images? Are these age-matched from time at egg lay?

The different panels show representative images from overnight embryo collections. This has been changed in the legend.

C: What does "embryos" mean? Same question for Fig 4A.

In this figure, embryos mean the exact number of embryos used to form the lysate for the western blot. The legend has been changed to address this.

Fig 4:

A: What does "embryos" mean? Number of embryos? Age in hours?

In this figure, embryos mean the exact number of embryos used to form the lysate for the western blot. The legend has been changed to address this.

C: Not sure the workflow figure panel is necessary, as I can't tell what each step does. This is better explained in methods. However I appreciated the short explanation in the text (lines 314-5).

The workflow panel helps to identify the samples labelled as input and aggregate for the western blot analysis. Since our input in the western blots does not refer to the total protein lysate, we feel it is helpful to point out exactly what stage at the protocol we are utilizing the sample for our analysis.

Minor comments:

The authors should describe the nature of the NASP alleles in the main text and present evidence of robust NASP depletion, potentially both in ovaries and in embryos. The antibody works well for westerns (Fig S2B). This is sort of demonstrated later in Figure 4A, but only in NAAP x twine activated eggs.

We appreciate the reviewer's comments about the NASP mutant allele. In our previous publication, we characterized the *NASP* mutant fly line and its effect on both stage 14 egg chambers and the embryos. We have revised the text to make this clearer.

Lines 163, 251, 339: minor typos

Line 184: It would help to clarify- I'm assuming cytoplasmic concentration (or overall) rather than nuclear concentration. If nuclear, I'd expect the opposite relationship. This occurs again when discussing NASP (line 267). I suspect it's also not absolute concentration, but relative concentration difference between cytoplasm and nucleus. It would help clarify if the authors were more precise.

We appreciate the reviewer's point and we have made the clarification to the text.

Line 189: Given that the "established integrative model" helps to reject the hypothesis that NASP is involved in H3 import, I think it's important to describe the model a little more, even though it's previously published.

We have revised the text to make this clearer.

Line 203: "The measured rate of H3.2 export from the nucleus is negligible" clarify this is in WT situations and not a conclusion from this study.

We have made the clarification in the text and the study has been referenced.

Line 211: How can the authors be so sure that the decrease in WT is due to "the loss of non-chromatin bound nucleoplasmic H3.2-Dendra2?"

From the live imaging experiments, the H3.2-Dendra2 intensity in the nucleus reduces dramatically upon nuclear envelope breakdown with the only H3.2-Dendra2 intensity remaining being the chromatin bound H3.2. Since a lot of H3.2 is imported into the nucleus and not all of it is incorporated into the chromatin, we mention that the intensity reduction is due to the loss of non-chromatin bound nucleoplasmic H3.2.

Line 217: In the conclusion, the authors indicate that NASP indirectly affects soluble supply of H3 in the nucleoplasm. I do believe they've shown that the import rate effect is indirect, but I don't know why they conclude that the effect of NASP on the soluble nucleoplasmic H3 supply is indirect. Similarly, the conclusion is indirect on line 239. Yet, the authors have not shown it's not direct, just assumed since NASP results in 50% decrease to deposited maternal histones.

We appreciate the feedback on the conclusions of Figure 2 from the reviewer. Our conclusions are primarily based on the effect of H3 levels in the absence of NASP in the early embryos. To establish direct causal effects, it would be important to recover the phenotypes by complementation experiments and providing molecular interactions to cause the effects. In this study we have not established those specific details to make conclusions of direct effects.

Line 292: What is the nature of the NASP "mutant?" Is it a null? Similarly, what kind of "mutant" is the *twine* allele? Line 295.

We have included descriptions of the *NASP* and *twine* mutants in the text. They are both null mutants.

Line 316: Why did the authors use stage 14 egg chambers here when they previously used embryos? This becomes more clear later shortly, when the authors examine activated eggs, but it's confusing in text.

The reason to use stage 14 egg chambers was to establish NASP function during oogenesis. We have modified the text to emphasize the reason behind using stage 14 egg chambers.

Lines 343-348: It's unclear if the authors are drawing extended conclusions here or if they are drawing from prior literature (if so, citations would be required). For example, why during oogenesis/embryogenesis are aggregation and degradation developmentally separated?

This conclusion is based primarily based on the findings from this study (Figure 4 and 5) and our previous published work. We have modified the text for more clarity.

Lines 386-7: I do not understand why the authors conclude that H3 aggregation and degradation are "developmentally uncoupled" and why, in the absence of NASP, "H3 aggregation precedes degradation."

This is based data in Figure 4 combined with our previous working showing that the total level of H3 is not changed in NASP-mutant stage 14 egg chambers. Aggregates seem to be more persistent in the stage 14 egg chambers (oogenesis) and they get cleared out upon egg activation (entry into embryogenesis). This provides evidence for aggregation occurring prior to degradation and these two events occurring in different developmental stages. Text changes have been made to clarify this.

Line 395: Why suddenly propose that NASP also functions in the nucleus to prevent aggregation, when earlier the authors suggest it functions only in the cytoplasm?

While we have shown that NASP can function in the cytoplasm, it does not rule out the likely possibility that NASP can also function in the nucleus. Given there has been a discrepancy in the literature as to where NASP functions, we felt this was an important point to address. We edited the text to state the NASP can function in the cytoplasm.

Lines 409-413: The authors claim that histone deficiency likely does not cause the embryonic arrest seen in embryos from NASP mutant mothers. This is because H3 is reduced by 50% yet some embryos arrest long before they've depleted this supply. However, the authors also showed that H3 import rates are affected in these embryos due to lower H3 concentration. Since the early embryo cycles are so rapid, reduced H3 import rates could lead to early arrest, even though available H3 remains in the cytoplasm.

We thank the reviewer for their suggestion. This conclusion is based on the findings from the previous study from our lab which showed that the majority of the embryos laid by NASP mutant females get arrested in the very early nuclear cycles (<NC5) whereas our live imaging studies show that NASP mutant embryos are capable of progressing into later nuclear cycles even with significantly reduced H3 levels. Thus, we conclude that H3 reduction is not sufficient to explain the defects observed in NASP mutant embryos.

Reviewer #1 (Significance (Required)):

The significance of the work is conceptual, as NASP is known to function in H3 availability but the precise mechanism is elusive. This work represents a necessary advance, especially to show that NASP does not affect H3 import rates, nor does it chaperone H3 into the nucleus. However, the authors acknowledge that many questions remain. Foremost, why is NASP imported into the nucleus and what is its role there?

I believe this work will be of interest to those who focus on early animal development, but NASP may also represent a tool, as the authors conclude in their discussion, to reduce histone levels during development and examine nucleosome positioning. This may be of interest to those who work on chromatin accessibility and zygotic genome activation.

I am a genetics expert who works in *Drosophila* embryogenesis. I do not have the expertise to evaluate the aggregate methods presented in Figure 4.

Reviewer #2 (Evidence, reproducibility and clarity (Required)):

Summary:

This manuscript focuses on the role of the histone chaperone NASP in *Drosophila*. NASP is a chaperone specific to histone H3 that is conserved in mammals. Many aspects of the molecular mechanisms by which NASP selectively binds histone H3 have been revealed through biochemical studies. However, key aspects of NASP's *in vivo* roles remain unclear, including where in the cell NASP functions, and how it prevents H3 degradation. Through live imaging in the early *Drosophila* embryo, which possesses large amounts of soluble H3 protein, Das et al determine that NASP does not control nuclear import or export of H3.2 or H3.3. Instead, they find through differential centrifugation analysis that NASP functions in the cytoplasm to prevent H3 aggregation and hence its subsequent degradation.

Major Comments:

1. The protein aggregation assays raise several questions.
 - a. From a technical standpoint, it would be helpful to have a positive control to demonstrate that the assay is effective at detecting protein aggregates. I.e. a genotype that exhibits increased protein aggregation; this could be for a protein besides H3.

A common issue raised by all three reviewers was to more convincingly demonstrate that assay that we have used to isolate protein aggregates does, in fact, isolate protein aggregates. To verify this, we have performed the aggregate isolation assay using controls that are known to induce more protein aggregation.

We have performed the aggregate isolation assay on WT egg chambers with chemical that are known to induce aggregate formation as positive controls for aggregate formation (Canavanine

and Azetidine-2-carboxylic acid). Additionally, we performed the aggregate isolation assay on S2 cells with Canavanine treatment to further validate the assay. Treatment of egg chambers and S2 cells with these chemicals increased aggregate formation as expected. We have included these critical controls in a revised Figure 5 and Figure S4. We have also modified the text accordingly.

b. If NASP is not required to prevent H3 degradation in egg chambers, then why are H3 levels much lower in NASP input lanes relative to wild-type egg chambers in Fig 4D?

We appreciate the reviewer's inputs regarding the reduced H3 levels in the NASP mutant egg chambers. We observe this reduction in H3 levels in the input because of the altered solubility of H3 which leads to the loss of H3 protein at different steps of the aggregate isolation assay. We have added a supplement figure showing H3 levels at different steps of the aggregate isolation assay (Fig. S4E). We do want to stress, however, that the total levels of H3 in stage 14 egg chambers does not change between WT and the *NASP* mutant. We have added a western blot in the supplemental figure to show that total H3 levels do not change in stage 14 egg chambers in WT and *NASP* female flies.

c. A corollary to this is that the increased fraction of H3 in aggregates in NASP mutants seems to be entirely due to the reduction in total H3 levels rather than an increase in aggregated H3. If NASP's role is to prevent aggregation in the cytoplasm, and degradation has not yet begun in egg chambers, then why are aggregated H3 levels not increased in NASP mutants relative to wild-type egg chambers? If the same number of egg chambers were used, shouldn't the total amount of histone be the same in the absence of degradation?

In previously published work, we demonstrated that total H3 levels are unaffected when comparing WT and *NASP* mutant stage 14 egg chambers. This means that the amount of H3 deposited into the eggs does not change in the absence of NASP. To address the reviewer's comment, we have included new western blot data showing that the total amount of H3 is unchanged between wild-type and *NASP*-mutant stage 14 egg chambers (revised Figure 4). We have also changed the text to make the link to our previous work clear.

2. The live imaging studies are well designed, executed, and quantified. They use an established genotype (H3.2-Dendra2) in wild-type and *NASP* maternal mutants to demonstrate that *NASP* is not directly involved in nuclear import of H3.2. Decreased import is likely due to reduced H3.2 levels in *NASP* mutants rather than reduced import rates per se. The same methodology was used to determine that loss of *NASP* did not affect H3.2 nuclear export. These findings eliminate H3.2 nuclear import/export regulation as possible roles for *NASP*, which had been previously proposed.

Thank you.

3. Live imaging also conclusively demonstrates that the levels of H3.2 in the nucleoplasm and in mitotic chromatin are significantly lower in NASP mutants than wild-type nuclei. Despite these lower histone levels, the nuclear cycle duration is only modestly lengthened.
4. The live imaging of NASP-Dendra2 nuclear import conclusively demonstrate that NASP and H3.2 are unlikely to be imported into the nucleus as one complex.

Thank you.

Minor Comments:

1. Additional details on how the NASP-Dendra2 CRISPR allele was generated should be provided. In addition, additional details on how it was determined that this allele is functional should be provided (e.g. quantitative assays for fertility/embryo viability of NASP-Dendra2 females)

Homozygous *NASP-Dendra2* flies are viable and fertile. This has now been mentioned in the text.

2. If statistical tests are used to determine significance, the type of test used should be reported in the figure legends throughout.

We have changed the text in the figure legend legends to make this clear.

3. The western blot shown in Figure 4A looks more like a 4-fold reduction in H3 levels in NASP mutants relative to wild-type embryos, rather than the quantified 2-fold reduction. Perhaps a more representative blot can be shown.

We have additional blots in the supplemental Figure S3C. The quantification was performed after normalization to the total protein levels and we have highlighted that in the figure legend.

Reviewer #2 (Significance (Required)):

As a fly chromatin biologist with colleagues that utilize mammalian experimental systems, I feel this manuscript will be of broad interest to the chromatin research community. Packaging of the genome into chromatin affects nearly every DNA-templated process, making the mechanisms by which histone proteins are expressed, chaperoned, and deposited into chromatin of high importance to the field. The study has multiple strengths, including high-quality quantitative imaging, use of a terrific experimental system (storage and deposition of soluble histones in early fly embryos). The study also answers outstanding questions in the field, specifically that NASP does not control nuclear import/export of histone H3. Instead, the authors propose that NASP functions to prevent protein aggregation. If this could be conclusively demonstrated, it would be valuable to the field. However, the protein aggregation studies need improvement. Technical demonstration that their differential centrifugation assay accurately detects aggregated proteins is needed. Further, NASP mutants do not exhibit increased H3 protein aggregation in the data presented. Instead, the increased fraction of aggregated H3 in NASP

mutants seems to be due to a reduction in the overall levels of H3 protein, which is contrary to the model presented in this paper.

Reviewer #3 (Evidence, reproducibility and clarity (Required)):

This manuscript by Das et al. entitled "NASP functions in the cytoplasm to prevent histone H3 aggregation during early embryogenesis", explores the role of the histone chaperone NASP in regulating histone H3 dynamics during early *Drosophila* embryogenesis. Using primarily live imaging approaches, the authors found that NASP is not directly involved in the import or export of H3. Moreover, the authors claimed that NASP prevents H3 aggregation rather than protects against degradation.

Major Comments:

Figure 1A-B: The plotted data appear to have substantial dispersion. Could the authors include individual data points or provide representative images to help the reader assess variability?

We chose to show unnormalized data in Figure 1 so readers could better compare the actual import values of H3 in the presence and absence of NASP. We felt it was a better representation of the true biological difference although raw data is more dispersive. We did also include normalized data in the supplement. Regardless, we have included supplemental movies of the H3.2-Dendra2 and NASP-Dendra2 to provide a visual reference of nuclear import of the two proteins. Since, each frame is a multi-z-stack and images are taken over 240s time frame, it's hard to provide a still representation of the import analysis. The raw data for each of our embryos containing all quantitative data is available as excel spreadsheets. Images will be made available upon reasonable request.

Given that the authors conclude that the reduced nuclear import is due to lowered H3 levels in NASP-deficient embryos, would overexpression of H3 rescue this phenotype? This would directly test whether H3 levels, rather than import machinery per se, drive the effect.

We thank the reviewer for their valuable suggestion. We and others have tried to overexpress histones in the *Drosophila* early embryo without success. There must be an undefined feedback mechanism preventing histone overexpression in the germline. In fact, a recent paper has been published in NSBM (PMID: 41491041) that indicates H4 protein could provide a feedback mechanism to prevent histone overexpression. While we would love to do this experiment, it is not technically feasible at this time.

Figure 2A-B: The authors present the Relative Intensity of H3-Dendra2, but this metric obscures absolute differences between Control and NASP knockout embryos. Please include Total Intensity plots to show the actual reduction in H3 levels.

This has been added to Figure S1J-K.

Additionally, Western blot analysis of nucleoplasmic H3 from wild-type vs. *NASP*-deficient embryos would provide essential biochemical confirmation of H3 level reductions.

We attempted a western of nuclei from 0-2 hr embryos laid by WT and *NASP* mutant flies for the validation of reduced nucleoplasmic H3 levels. While this may seem like a simple experiment, it turned out to be more complex and difficult to quantify than we anticipated.

To get enough embryos to perform a nuclei isolation and western blot we performed 0-2h collections. The problem is that the number of nuclei in embryos laid by wild-type mothers and *NASP*-mutant mothers are considerably different given that ~50% of embryos collected from *NASP*-mutant mothers arrest with only one or two nuclei (Tirgar et al., 2023 *PLoS Genetics*). We have tried to use qPCR to normalize to DNA levels. However, given reduced nuclei numbers and reduced amount of H3 in embryos laid by *NASP*-mutant mothers, we would have to collect grams of embryos to be able to get a reliable signal to quantify. Even then, the comparison would be approximate. We have included a panel from a representative experiment for reference (see below). We just don't get enough H3 signal in the nuclei from *NASP* mutant embryos to quantify the difference.

The differences in nuclei number driven by developmental defects when comparing WT and *NASP*-mutant embryos highlights why quantitative imaging is the best way to measure the nucleoplasmic and chromatin bound levels of H3.

Figure 4: To support the conclusion that *NASP* prevents H3 aggregation, I recommend performing aggregation assays by adding compounds that induce unfolding (amino acid analogues that induce unfolding, like canavanine or Azetidine-2-carboxylic acid) or using aggregation-prone H3 mutants.

We thank the reviewer for this fantastic suggestion! We have performed the aggregate isolation assay on WT egg chambers with chemical that are known to induce aggregate formation as positive controls for aggregate formation (canavanine and azetidine-2-carboxylic acid). Additionally, we performed the aggregate isolation assay on S2 cells with canavanine treatment to further validate the assay. Treatment of egg chambers and S2 cells with these chemicals increased aggregate formation as expected. We have included these critical controls in a revised Figure 5 and Figure S4. We have also modified the text accordingly.

Inclusion of CMA and proteasome inhibition experiments could also clarify whether degradation pathways are secondarily involved or compensatory in the absence of NASP.

The degradation pathway for H3 in the absence of NASP is unknown and a major focus of our future work is to define this pathway. *Drosophila* does not have a CMA pathway and therefore, we don't know how H3 aggregates are being sensed.

Minor Comments:

(1) The Introduction would benefit from mentioning the two NASP isoforms that exist in mammals (sNASP and tNASP), as this evolutionary context may inform interpretation of the *Drosophila* results.

We added this to the introduction.

(2) Could the authors comment on the status of histone H4 in their experimental system? Given the observed cytoplasmic pool of H3, is it likely to exist as a monomer? If this H3 pool is monomeric, does that suggest an early failure in H3-H4 dimerization, and could this contribute to its aggregation propensity?

In our previous work we noted that NASP binds more preferentially to H3 and the levels of H3 we much more reduced upon NASP depletion than H4. We pointed out in this publication that our data was consistent with H3 stores being monomeric in the *Drosophila* embryo. We don't have a H4-Dendra2 line to test. In the future, however, this is something we are very keen to look at.

Reviewer #3 (Significance (Required)):

This work addresses a timely and important question in the field of chromatin biology and developmental epigenetics. The focus on histone homeostasis during embryogenesis and the cytoplasmic role of NASP adds a novel perspective. The live imaging experiments are a clear strength, providing valuable spatiotemporal insights. However, I believe that the manuscript would benefit significantly from additional biochemical validation to support and clarify some of the mechanistic claims.

March 24, 2026

RE: JCB Manuscript #202511182R

Jared Nordman
Vanderbilt University

Dear Dr. Nordman:

Thank you for submitting your revised manuscript entitled "NASP functions in the cytoplasm to prevent histone H3 aggregation during early embryogenesis". The reviewers all now support publication so we would be happy to publish your paper in JCB pending final revisions necessary to meet our formatting guidelines (see details below).

In your final revision, please be sure to address reviewer #3's final minor concerns.

A. MANUSCRIPT ORGANIZATION AND FORMATTING:

Full guidelines are available on our Instructions for Authors page, <http://jcb.rupress.org/submission-guidelines#revised>.

1) Text limits: Character count for Articles is < 40,000, not including spaces. Count includes abstract, introduction, results, discussion, and acknowledgments. Count does not include title page, figure legends, materials and methods, references, tables, or supplemental legends.

2) Figures limits: Articles may have up to 10 main text figures.

****3) Figure formatting:**

****Scale bars must be present on all microscopy images, including inset magnifications.**

****Molecular weight or nucleic acid size markers must be included on all gel electrophoresis (e.g. Fig. 4, 5, S3, S4 and S5).**

Aspect ratios of images may not be altered.

4) Statistical analysis: Error bars on graphic representations of numerical data must be clearly described in the figure legend. The number of independent data points (n) represented in a graph must be indicated in the legend. Statistical methods should be explained in full in the materials and methods. For figures presenting pooled data the statistical measure should be defined in the figure legends. Please also be sure to indicate the statistical tests used in each of your experiments (either in the figure legend itself or in a separate methods section) as well as the parameters of the test (for example, if you ran a t-test, please indicate if it was one- or two-sided, etc.). Also, if you used parametric tests, please indicate if the data distribution was tested for normality (and if so, how). If not, you must state something to the effect that "Data distribution was assumed to be normal but this was not formally tested."

5) Abstract and title: The abstract should be no longer than 160 words and should communicate the significance of the paper for a general audience. The title should be less than 100 characters including spaces. Make the title concise but accessible to a general readership.

6) Materials and methods: Should be comprehensive and not simply reference a previous publication for details on how an experiment was performed. Please provide full descriptions in the text for readers who may not have access to referenced manuscripts.

****7) All antibodies, cell lines, animals, and tools used in the manuscript should be described in full, including accession numbers for materials available in a public repository such as the Resource Identification Portal. Please be sure to provide the sequences for all of your primers/oligos and RNAi constructs in the materials and methods. You must also indicate in the methods the source, species, and catalog numbers (where appropriate) for all of your antibodies. Please also indicate the acquisition and quantification methods for immunoblotting/western blots.**

8) Microscope image acquisition: The following information must be provided about the acquisition and processing of images:

- a. Make and model of microscope
- b. Type, magnification, and numerical aperture of the objective lenses
- c. Temperature
- d. Imaging medium
- e. Fluorochromes

- f. Camera make and model
 - g. Acquisition software
 - h. Any software used for image processing subsequent to data acquisition. Please include details and types of operations involved (e.g., type of deconvolution, 3D reconstitutions, surface or volume rendering, gamma adjustments, etc.).
- 9) References: There is no limit to the number of references cited in a manuscript. References should be cited parenthetically in the text by author and year of publication. Abbreviate the names of journals according to PubMed.
- 10) Supplemental materials: There are strict limits on the allowable amount of supplemental data. Articles may have up to 5 supplemental figures. Please also note that tables, like figures, should be provided as individual, editable files. A summary of all supplemental material should appear at the end of the Materials and methods section.
- 11) eTOC summary: A ~40-50-word summary that describes the context and significance of the findings for a general readership should be included on the title page. The statement should be written in the present tense and refer to the work in the third person.
- 12) Conflict of interest statement: JCB requires inclusion of a statement in the acknowledgements regarding competing financial interests. If no competing financial interests exist, please include the following statement: "The authors declare no competing financial interests." If competing interests are declared, please follow your statement of these competing interests with the following statement: "The authors declare no further competing financial interests."
- **13) ORCID IDs: ORCID IDs are unique identifiers allowing researchers to create a record of their various scholarly contributions in a single place. Please note that ORCID IDs are now *required* for all authors. At resubmission of your final files, please be sure to provide your ORCID ID and those of all co-authors.
- 14) A separate author contribution section following the Acknowledgments. All authors should be mentioned and designated by their full names. We encourage use of the CRediT nomenclature.

Please note that JCB now requires authors to submit Source Data used to generate figures containing gels and Western blots with all revised manuscripts. This Source Data consists of fully uncropped and unprocessed images for each gel/blot displayed in the main and supplemental figures. For assays performed using capillary electrophoresis and/or immunoassay-based detection, authors should instead provide the electropherogram graph(s) for each experiment, plotting fluorescence/chemiluminescence intensity vs. molecular weight/size. Please be sure to provide one Source Data file for each figure gels, blots, and/or capillary electrophoresis assays along with your revised manuscript files. File names for Source Data figures should be alphanumeric without any spaces or special characters (i.e., SourceDataF#, where F# refers to the associated main figure number or SourceDataFS# for those associated with Supplementary figures). For traditional gels and blots, the lanes of the gels/blots should be labeled as they are in the associated figure, the place where cropping was applied should be marked (with a box), and molecular weight/size standards should be labeled wherever possible. For capillary electrophoresis assays, each trace in the graph should be color-coded and labeled to indicate which protein, gene, or sample is being measured (please try to avoid red/green combinations to accommodate our color-blind readers).

Journal of Cell Biology now requires a data availability statement for all research article submissions. These statements will be published in the article directly above the Acknowledgments. The statement should address all data underlying the research presented in the manuscript. Please visit the JCB instructions for authors for guidelines and examples of statements at (<https://rupress.org/jcb/pages/editorial-policies#data-availability-statement>).

B. FINAL FILES:

The license to publish form must be signed before your manuscript can be sent to production. A link to the license to publish form will be sent to the corresponding author only. Please take a moment to check your funder requirements before choosing the appropriate license.

Thank you for your attention to these final processing requirements. Please revise and format the manuscript and upload materials within 7 days. If you need an extension for whatever reason, please let us know and we can work with you to determine a suitable revision period.

Thank you for this interesting contribution, we look forward to publishing your paper in Journal of Cell Biology.

Sincerely,

Hiroshi Kimura
Monitoring Editor
Journal of Cell Biology

Gabriele Stephan
Scientific Editor
Journal of Cell Biology

Reviewer #1:

The authors investigate the function of the H3 chaperone NASP, which is known to bind directly to H3 and prevent degradation of soluble H3. What is unclear is where NASP functions in the cell (nucleus or cytoplasm), how NASP protects H3 from degradation (direct or indirect), and if NASP affects H3 dynamics (nuclear import or export). They use the powerful model system of *Drosophila* embryos because the soluble H3 pool is high due to maternal deposition and they make use of photoconvertible Dendra-tagged proteins, since these are maternally deposited and can be used to measure nuclear import/export rates.

Using these systems and tools, they conclude that NASP affects nuclear import, but only indirectly, because embryos from NASP mutant mothers start out with 50% of the maternally deposited H3. Because of the depleted H3 and reduced import rates, NASP deficient embryos also have reduced nucleoplasmic and chromatin-associated H3. Using a new Dendra-tagged NASP allele, the authors show that NASP and H3 have different nuclear import rates, indicating that NASP is not a chaperone that shuttles H3 into the nucleus. They test H3 levels in embryos that have no nuclei and conclude that NASP functions in the cytoplasm, and through protein aggregation assays they conclude that NASP prevents H3 aggregation.

The revised data, notably the aggregate data with controls, are supportive of the conclusions.

The authors have put substantial work into improving the initial submission. I do not advise any additional revisions.

Reviewer #2:

The authors have carefully addressed all the comments I raised. I believe the manuscript is now suitable for publication.

Reviewer #3:

This manuscript focuses on the role of the histone chaperone NASP in *Drosophila*. NASP is a chaperone specific to histone H3 that is conserved in mammals. Many aspects of the molecular mechanisms by which NASP selectively binds histone H3 have been revealed through biochemical studies. However, key aspects of NASP's *in vivo* roles remain unclear, including where in the cell NASP functions, and how it prevents H3 degradation. Through live imaging in the early *Drosophila* embryo, which possesses large amounts of soluble H3 protein, Das et al determine that NASP does not control nuclear import or export of H3.2 or H3.3. Instead, they find through differential centrifugation analysis that NASP functions in the cytoplasm to prevent H3 aggregation and hence its subsequent degradation.

The authors have significantly strengthened the paper's findings relative to the initial manuscript version in Review Commons by incorporating controls for the protein aggregation assay.

My only remaining comments involve requested changes to the text.

1. The authors should explain in the Results section their interpretation for the decreased histone H3 Input signal in Figure 5D.
2. Although they cover the topic of degradation in the discussion, the authors should make it more clear in the Results section that histone H3 protein degradation is an inference, and the pathway that degrades aggregated H3 in NASP mutants has not yet been identified.
3. line 391, it is too strong of a claim based on the data presented to state that NASP "primarily functions in the cytoplasm"
4. Line 167 and elsewhere, it is not clear that the Horard and Loppin, 2015 review article provides sufficient evidence for the accompanying statement that >99% of maternally deposited histones are soluble in early embryos.